# Null infinity as an inverted extremal horizon: Matching an infinite set of conserved quantities for gravitational perturbations

Shreyansh Agrawal[1,2]*, Panagiotis Charalambous[1,2]† and Laura Donnay[1,2]‡

**1** International School for Advanced Studies (SISSA),
Via Bonomea 265, 34136 Trieste, Italy
**2** National Institute for Nuclear Physics (INFN),
Sezione di Trieste, Via Valerio 2, 34127, Italy

★ sagrawal@sissa.it ,    † pcharala@sissa.it ,    ‡ ldonnay@sissa.it ,

## Abstract

Every spacetime that is asymptotically flat near null infinity can be conformally mapped via a spatial inversion onto the geometry around an extremal, non-rotating and non-expanding horizon. We set up a dictionary for this geometric duality, connecting the geometry and physics near null infinity to those near the dual horizon. We then study its physical implications for conserved quantities for extremal black holes, extending previously known results to the case of gravitational perturbations. In particular, we derive a tower of near-horizon gravitational charges that are exactly conserved and show their one-to-one matching with Newman-Penrose conserved quantities associated with gravitational perturbations of the extremal Reissner-Nordström black hole geometry. We furthermore demonstrate the physical relevance of spatial inversions for extremal Kerr-Newman black holes, even if the latter are notoriously not conformally isometric under such inversions.

## Contents

---

# 1 Introduction

The study of null surfaces has proven paramount in understanding gravitational physics. A very well-known class of such surfaces that is characteristic to asymptotically flat spacetimes are the surfaces where radiation comes from and reaches at large distances, i.e. past and future null infinities, $\mathscr{I}^-$ and $\mathscr{I}^+$. The asymptotic structure of gravity near null infinity has been instrumental in proving the existence of gravitational radiation within the full non-linear regime of Einstein's general theory of relativity [1–6]. This concrete theoretical prediction has been confirmed by the triumphant detection of the GW150914 signal by the LIGO interferometers [7]. Today, gravitational waves are routinely observed through advanced interferometric apparatuses of the LIGO-VIRGO-KAGRA collaboration, with the current third Gravitational-Wave Transient Catalog (GWTC-3) reporting 90 confirmed detections of transient gravitational waves emitted during the coalescence of binary systems of compact bodies [8]. Of those, 83 consist of signals from the coalescence of binary systems of black holes, objects which are equipped with another fundamental type of null surfaces: black hole event horizons.

Both future null infinity and the black hole (future) event horizon are classically well-defined properties of an asymptotically flat geometry; one is generated by asymptotic outgoing null rays, while the other is the past null cone of the former.[1] Since both geometries are by definition null hypersurfaces, they naturally inherit a 'Carrollian' structure [15–22], characterized by a degenerate metric. For black holes, this has been explicitly demonstrated in Refs. [23,24] and further studied in Refs. [25–29].

Recently, there has been growing interest utilizing structural similarities between these two types of geometries [26, 29–36]. Notably, Refs. [31–34] incorporated null infinity into the framework of non-expanding horizons, while Ref. [35] examined a relationship between

---

[1]This definition pertains to absolute/causal horizons. In the generic setup, the location of the event horizon requires a *global* definition and is inherently teleological. For the special case of stationary black holes, however, the event horizon is a Killing horizon and can, thus, be locally defined by the Killing vector that generates it. Alternative local definitions that asymptote to the event horizons of stationary black holes at late times rely on concepts such as that of apparent horizons [9] and trapping horizons [10], see also Refs. [11–14] and references therein for more information.

stretched horizons and asymptotic infinity. Despite these shared structural features, their respective physical properties remain fundamentally distinct. For instance, the boundary structure of null infinity is universal and its phase space [37–39] comprises of a hard sector, containing non-radiative data such as multipole moments [40], as well as a soft sector that is associated with the vacuum structure [39, 41–43]. In contrast, the phase space associated with a null surface at a finite distance, such as a black hole horizon, describes a fluctuating boundary [28, 36, 44–56].

These differences extend to their asymptotic symmetries. At future null infinity, the asymptotic isometry group is spanned by the BMS group [3,5,6]. Its characteristic enhancement compared to the Poincaré group is the existence of supertranslations, angle-dependent translations of the retarded time. At the horizon $\mathscr{H}^+$, one instead finds an infinite extension of supertranslations with unrestricted dependence on the advanced time, precisely due to the fact that the near-horizon surface gravity and boundary metric are fluctuating free data [44–46]. In the case of extremal horizons, these reduce to supertranslations and superdilatations[2]. From this symmetry perspective, it is therefore natural to seek a connection between the geometry of an asymptotically flat spacetime and the geometry near an extremal horizon whose horizon metric is fixed.

A notable classical property of extremal horizons is their dynamical instability under linearized perturbations. This was first demonstrated by Aretakis in Refs. [57–59] for the case of scalar perturbations of stationary and axisymmetric extremal horizons, and was soon generalized to electromagnetic and linearized gravitational perturbations of extremal Kerr and Reissner-Nordström black holes [60, 61].[3] More precisely, Aretakis studied solutions to the wave equation on extremal black holes and showed that, for generic initial data on a spacelike surface intersecting the future horizon, first order derivatives of the scalar field transverse to the future event horizon $\mathscr{H}^+$ do not decay along $\mathscr{H}^+$, while higher-order derivatives in fact blow up at late advanced time $\upsilon \to \infty$. Remarkably, these instabilities only depend on the *local* geometric properties of the horizon and result from the existence of an infinite hierarchy of *conservation laws* along extremal horizons.

The origin of this infinite tower of Aretakis conserved quantities is not well understood: while they are sometimes referred to as 'Aretakis charges'[4], it is still unknown whether they can be interpreted as Noether charges associated to some type of near-horizon (or perhaps more hidden) symmetries. In fact, their extraction from an expansion of the equations of motion is reminiscent of the derivation of another set of mysterious 'charges': the linearly and non-linearly Newman-Penrose conserved quantities that arise from a near–null infinity asymptotic expansion of the equations of motion [65–67]. As emphasized in Ref. [61], Aretakis' conserved quantities are related to outgoing radiation at $\mathscr{H}^+$, while the Newman-Penrose constants are closely related to incoming radiation at $\mathscr{I}^+$. In fact, there exists a precise relationship between the near-horizon Aretakis charges and the Newman-Penrose constants at null infinity, as first observed in Refs. [61, 68] (and further studied in Refs. [69–75]).

In this work, we study further the relation between the surfaces of null infinity and horizon at a finite distance. The structure of our paper is the following. In Section 2, we review the physics near each of the two null surfaces. After highlighting their different dynamics, we show the existence of discrete spatial inversions that conformally map the geometry of an asymp-

---

[2]This is to be contrasted with the fact that null infinity is a conformal boundary equipped with a non-fluctuating boundary metric which completely fixes these would-be superdilatations.

[3]In Ref. [61] and, more recently in Ref. [62], the instability of the electrically charged extremal Reissner-Nordström black hole under coupled electromagnetic and gravitational perturbations was studied. See also Ref. [63] and Refs. [61, 64] for generalizations to massive scalar fields and to higher dimensional extremal black hole geometries.

[4]We will sometimes also adopt this name in what follows, bearing in mind that this could not be the most appropriate nomenclature.

totically flat spacetime to the geometry near an extremal, non-expanding and non-twisting horizon[5]. This allows the construction of a dictionary that relates geometric quantities living on one null surface to quantities living on the corresponding dual-under-spatial-inversion null surface. We also review and contrast the symmetries that preserve the structure near null infinity and near the horizon. As we clarify there, these naturally arise as subsectors of the larger class of Carrollian conformal symmetries.

In Section 3, we consider the explicit example of the four-dimensional extremal Reissner-Nordström black hole, which has the special property of being self-dual under the aforementioned type of spatial inversions. This property provides a geometric explanation for the existence of the well-known discrete conformal isometry of Couch & Torrence [76]. We analyze the equations of motion associated with scalar, electromagnetic (with frozen gravitational field) and gravitational (with constrained electromagnetic perturbations) perturbations of the extremal Reissner-Nordström black hole in a unified framework. This allows us to extract the infinite towers of near-horizon (Aretakis) charges and the near-null infinity (Newman-Penrose) charges for any spin-weight-$s$ perturbations. We then apply our analysis to extract physical constraints, namely, we demonstrate the one-to-one matching between infinite hierarchy of Aretakis and Newman-Penrose conserved quantities, extending previous results [61,68–71,74,75] to the more intricate case of gravitational perturbations. As we explain, the key feature that allows us to derive the analogs of these results for the gravitational case is the fact that the spin-weighted wave operator of extremal Reissner-Nordström is conformally invariant under conformal inversions.

In Section 4, we demonstrate the physical relevance of this type of spatial inversions even in situations where the near-horizon geometry is not dual to an asymptotically flat spacetime. Namely, we study spin-weighted perturbations of the extremal Kerr-Newman black hole and reveal the existence of phase space spatial inversions that are conformal symmetries of the equations of motion, extending the results of Ref. [76] beyond the scalar perturbations paradigm. After extracting the Newman-Penrose and Aretakis conserved quantities associated with axisymmetric spin-weighted perturbations of the rotating black hole, we demonstrate that this sector of perturbations inherits a geometric spatial inversion conformal symmetry which precisely imposes the matching of these charges.

We finish with a discussion of our results and various future directions in Section 5. We also supplement with Appendix A, reviewing some basic elements of the Newman-Penrose formalism needed for performing the calculations, and Appendix B, collecting expressions of the Newman-Penrose and Aretakis charges that display the explicit mixing of spherical harmonic modes induced by the non-zero angular momentum of the Kerr-Newman black hole.

*Notation and conventions*: In this work, we employ geometrized units with the speed of light and Newton's gravitational constant set to unity, $c = G_\mathrm{N} = 1$, and we adopt the mostly-positive metric Lorentzian signature. Spacetime indices will be denoted by small Latin indices from the beginning of the alphabet, e.g. $a$, $b$, $c$, ranging from 0 to $d-1$, for a $(1+(d-1))$-dimensional spacetime, while capital Latin indices from the beginning of the alphabet, e.g. $A$, $B$, $C$, will denote angular directions transverse to null surfaces, ranging from 1 to $d-2$. We will refer to such transverse directions as "spatial" directions, with the corresponding intrinsic metric describing the geometry of the submanifolds spanned by such spatial coordinates dubbed the "spatial" metric. Repeated indices will be summed over. The symbol $\hat{=}$ will be used to denote "equality on the null surface". The metric on $\mathbb{S}^{d-2}$ and its inverse will be denoted by $\gamma_{AB}$ and $\gamma^{AB}$ respectively.

---

[5]As explained in more details later, this conformal isomorphism arises from the realization that $\mathscr{I}$ is also a non-expanding horizon that is furthermore extremal and non-twisting, thanks to the existence of preferred divergence-free conformal frames [32,33].

## 2    Duality between null infinity and extremal horizon

In this section, we motivate a correspondence between null infinity ($\mathscr{I}$) and a horizon ($\mathscr{H}$) that is extremal, non-expanding and non-twisting, and set up a $\mathscr{I}/\mathscr{H}$ dictionary between geometric quantities associated with each of the null surfaces. We begin with a review of the asymptotic expansions relevant for each of the two null surfaces and present their geometric properties by means of evolution and hypersurface equations. We then remark a mapping from one null surface to the other under spatial inversions and uncover a duality between them. In doing so, we also point out the inequivalent physics at the two null surfaces [32, 33], as well as review the asymptotic symmetries manifesting at each null surface.

### 2.1    Geometry near a finite-distance horizon

Let us start with the horizon side of the correspondence. We refer the reader to Ref. [77] for a clear review of horizon geometry. Here, we will adopt null Gaussian coordinates $(v, \rho, x^A)$ constructed as prescribed e.g. in Ref. [14]. Namely, the coordinate $v$ is chosen to be an advanced time coordinate, whose level-sets are null surfaces, the radial coordinate $\rho$ is chosen to be an affine parameter of the generators of these null surfaces, and the remaining transverse coordinates $x^A$, $A = 1, \ldots, d-2$, henceforth referred to as "spatial coordinates", are chosen to be constant on each such null generator. At the level of the metric, these light-cone gauge conditions set $g^{vv} = 0$, $g^{v\rho} = +1$ and $g^{vA} = 0$ respectively, or, equivalently, $g_{\rho\rho} = 0$, $g_{v\rho} = +1$ and $g_{\rho A} = 0$. This gauge fixing is analogous to the Newman-Unti gauge at infinity [78].

Setting the (future) horizon $\mathscr{H}^+$ at the $\rho = 0$ null surface, the spacetime is described by the line element [14, 79]

$$ds^2_{\mathscr{H}^+} = -\rho^2 \mathcal{F} dv^2 + 2\, dv d\rho + g_{AB} \big( dx^A + \rho\, \theta^A dv \big) \big( dx^B + \rho\, \theta^B dv \big)\,, \tag{2.1}$$

while the corresponding inverse metric can be read from

$$\partial^2_{\mathscr{H}^+} = \rho^2 \mathcal{F} \partial^2_\rho + 2\, \partial_v \partial_\rho - 2\rho\, \theta^A \partial_\rho \partial_A + g^{AB} \partial_A \partial_B\,, \tag{2.2}$$

with $g^{AB}$ the components of the inverse of the spatial metric $g_{AB}$.

On top of the light-cone gauge conditions, we impose the following near-horizon fall-offs of the various fields entering the metric [45]

$$
\begin{aligned}
\mathcal{F}\big(v, \rho, x^A\big) &= 2\rho^{-1} \kappa\big(v, x^A\big) + \mathcal{F}_0\big(v, x^A\big) + o\big(\rho^{+0}\big)\,, \\
\theta^A\big(v, \rho, x^B\big) &= \vartheta^A\big(v, x^B\big) + o\big(\rho^{+0}\big)\,, \\
g_{AB}\big(v, \rho, x^C\big) &= \Omega_{AB}\big(v, x^C\big) + \rho\, \lambda_{AB}\big(v, x^C\big) + o\big(\rho\big)\,.
\end{aligned}
\tag{2.3}
$$

To study the physics at the horizon, we introduce a null vector $\vec{\ell}$ and a null 1-form $n$ according to [24]

$$\vec{\ell} = \ell^a \partial_a = \partial_v - \rho\, \theta^A \partial_A + \frac{1}{2} \rho^2 \mathcal{F} \partial_\rho\,, \quad n = n_a dx^a = -dv\,. \tag{2.4}$$

These have been constructed such that $\ell_a \ell^a = n_a n^a = 0$ and $n_a \ell^a = -1$ *everywhere* and they are appropriate for studying the intrinsic geometry of a level-$v$ null surface. In particular, the outgoing null ray vector $\vec{\ell}$ coincides with the null normal at the horizon, $\vec{\ell} \,\hat{=}\, \partial_v$, while the ingoing null ray vector $\vec{n} = g^{ab} n_b \partial_a = -\partial_\rho$ is transverse to the horizon and aligned with the ingoing null geodesics everywhere in the exterior. Here, and in the rest of this work, the symbol " $\hat{=}$ " means "evaluated at $\rho = 0$".

Using these null vielbein vectors, the intrinsic spatial metric $q_{ab}$ of the $\rho = $ const. surface can be isolated via the bulk metric decomposition

$$g_{ab} = -2\ell_{(a}n_{b)} + q_{ab}.$$ (2.5)

In particular, the intrinsic metric becomes the spatial metric on the horizon,

$$\begin{aligned} q_{ab}dx^a dx^b &= g_{AB}\left(dx^A + \rho\,\theta^A dv\right)\left(dx^B + \rho\,\theta^B dv\right) \\ &\hat{=} 0 \cdot dv^2 + 0 \cdot dv dx^A + \Omega_{AB}dx^A dx^B. \end{aligned}$$ (2.6)

Such degenerate metrics are inherently endowed with a 'Carrollian' structure; see e.g. Refs. [15–22, 80–85].

The extrinsic geometry is captured by the longitudinal deformation tensor $\Sigma_{ab} = \frac{1}{2}q_a{}^c q_b{}^d \mathcal{L}_\ell q_{cd}$ (or second fundamental form), the twist (Hájíček) 1-form field $\omega_a = -q_a{}^b n_c \nabla_b \ell^c$, the non-affinity coefficient $\tilde{\kappa}$, defined via $\ell^b \nabla_b \ell^a = \tilde{\kappa}\ell^a$,[6] and the transversal deformation tensor $\Xi_{ab} = \frac{1}{2}q_a{}^c q_b{}^d \mathcal{L}_n q_{cd}$. On the horizon, the non-zero components are evaluated to be [24]

$$\Sigma_{AB} \hat{=} \frac{1}{2}\partial_v \Omega_{AB}, \quad \omega_A \hat{=} -\frac{1}{2}\vartheta_A, \quad \tilde{\kappa} \hat{=} \kappa, \quad \Xi_{AB} \hat{=} -\frac{1}{2}\lambda_{AB},$$ (2.7)

where spatial indices are lowered and raised using $\Omega_{AB}$ and its inverse, $\Omega^{AB}$. From the longitudinal deformation tensor, the expansion $\Theta = q^{ab}\Sigma_{ab}$ of the null normal $\vec{\ell}$ and the longitudinal shear $\sigma_{ab} = \Sigma_{ab} - \frac{1}{d-2}q_{ab}\Theta$ can be extracted to be [24]

$$\sigma_{AB} \hat{=} \frac{1}{2}\partial_v \Omega_{AB} - \frac{1}{d-2}\Omega_{AB}\Theta, \quad \Theta \hat{=} \partial_v \ln\sqrt{\Omega},$$ (2.8)

with $\sqrt{\Omega} := \sqrt{\det(\Omega_{AB})}$ the volume element of the horizon spatial metric. Similarly, from the transversal deformation tensor, the expansion $\Theta^{(n)} = q^{ab}\Xi_{ab}$ of the ingoing null transverse vector $\vec{n}$ and the transversal shear $\sigma_{ab}^{(n)} = \Xi_{ab} - \frac{1}{d-2}q_{ab}\Theta^{(n)}$ can be extracted to be [77]

$$\sigma_{AB}^{(n)} \hat{=} -\frac{1}{2}\lambda_{\langle AB\rangle}, \quad \Theta^{(n)} \hat{=} -\frac{1}{2}\lambda_A{}^A,$$ (2.9)

where "$\langle AB\rangle$" means "the symmetric trace-free part with respect to the horizon spatial metric", e.g. $\lambda_{\langle AB\rangle} := \lambda_{(AB)} - \frac{1}{d-2}\Omega_{AB}\lambda_C{}^C$.

In summary, we see that the asymptotic fields entering the near-horizon expansion of the geometry acquire a very physical interpretation: $\kappa$ is the surface gravity, $\vartheta_A$ is the twist 1-form, $\Omega_{AB}$ is the horizon spatial metric, whose time dependence determines the longitudinal shear and the expansion of the null normal $\vec{\ell}$, and $\lambda_{AB}$ is the transversal deformation rate of the horizon.

At this point, let us clarify the role of the field $\kappa(v, x^A)$ that enters the near-$\mathcal{H}$ expansion of the metric and its distinction from the non-affinity coefficient $\tilde{\kappa}$ of the null vector generating the horizon. The non-affinity coefficient $\tilde{\kappa}$ is a scalar field defined intrinsically on the horizon,

---

[6]Even though the null tetrad vector $\vec{\ell}$ we chose here is the null normal on the horizon, it is not aligned with outgoing null geodesics everywhere, namely, $\ell^b \nabla_b \ell^a$ is not proportional to $\ell^a$ away from the horizon, unless $D_A \mathcal{F} = 0$. Nevertheless, this can be achieved for generic geometries by adding the following "far-horizon" correction

$$\vec{\ell} \rightarrow \vec{\tilde{\ell}} = \vec{\ell} + \rho\, L^A\left(\partial_A - \frac{1}{2}\rho\, L_A \partial_\rho\right)$$

where $L^A = L^A(v, \rho, x^B)$ is a transverse vector and $L_A := g_{AB}L^B$ here. For $\vec{\tilde{\ell}}$ to be geodesic, this transverse vector must satisfy an evolution equation which can be solved order by order in a near-horizon expansion. For instance, writing $L_A = L_A^{(0)} + \mathcal{O}(\rho)$, at leading order one needs to have $(\partial_v + 2\kappa)L_A^{(0)} = -D_A\kappa$. Then, the resulting vector field $\vec{\tilde{\ell}}$ is aligned *everywhere* with null geodesics that become outgoing on the horizon.

and, as the name suggests, it captures the information of whether the parameter $\tau$, associated with $\vec{\ell}$ by $\ell^a = \frac{dx^a}{d\tau}$, is an affine parameter of the null geodesics generating the horizon ($\tilde{\kappa} = 0$) or not ($\tilde{\kappa} \neq 0$). Its value can be freely chosen by scalings of the form $\vec{\ell} \to \alpha\vec{\ell}$, which preserve the structure of the horizon for any smooth and non-vanishing scalar field $\alpha$, since then $\tilde{\kappa} \to \alpha(\tilde{\kappa} + \nabla_\ell \ln \alpha)$. For instance, one can always choose $\tilde{\kappa} = 0$. On the other hand, the metric field $\kappa(v, x^A)$, that enters Eq. (2.1) according to Eq. (2.3), is independent of the choice of the null vector $\vec{\ell}$ and it is a property of the geometry, namely, a boundary condition for the behavior of the metric near the horizon.

Acknowledging the importance of the near-horizon boundary conditions, we will coin the transition from $g_{vv} = \mathcal{O}(\rho)$ to $g_{vv} = \mathcal{O}(\rho^2)$ the term "extremality", i.e., in the current work, an extremal horizon is one for which $\kappa(v, x^A) = 0$, regardless of what the value of the non-affinity coefficient $\tilde{\kappa}$ of $\vec{\ell}$ is. While this distinction between the metric field $\kappa$ and the non-affinity coefficient $\tilde{\kappa}$ is important, the null vector field $\vec{\ell}$ used here has been chosen such that $\tilde{\kappa}$ coincides with $\kappa(v, x^A)$ on the horizon, so as to be able to propagate the definition of extremality at the level of $\tilde{\kappa}$, but one should bare in mind that this definition is independent of the choice of $\vec{\ell}$.

**Horizon dynamics**    Let us now turn to the dynamics of these intrinsic objects. The evolution equations for the expansion $\Theta$, the twist 1-form $\omega_A$, and the transversal shear $\lambda_{AB}$ are governed by Einstein equations. The first and most famous one is the one governing the evolution of the expansion, known as the null Raychaudhuri equation[7] [86],

$$\ell^a \ell^b R_{ab} \triangleq -\left[(\partial_v - \kappa)\Theta + \frac{1}{d-2}\Theta^2 + \sigma_{AB}\sigma^{AB}\right]. \tag{2.10}$$

The twist evolution gives the Damour equation [87, 88],

$$q_a{}^b \ell^c R_{bc} \triangleq -\frac{1}{2}\delta_a^A\left[(\partial_v + \Theta)\vartheta_A + 2D_A\left(\kappa + \frac{d-3}{d-2}\Theta\right) - 2D^B \sigma_{AB}\right]. \tag{2.11}$$

While the above equation shares some resemblance with the Navier-Stokes equation for a viscous fluid, it was pointed out in Ref. [24] that Eqs. (2.10), (2.11) should rather be regarded as conservation equations of a *Carrollian* fluid [89, 90], rather than a Galilean one. Last, the dynamics of the transversal shear $\lambda_{AB}$ are governed by the 'transversal deformation rate evolution equation' (following the nomenclature of Ref. [77])

$$\begin{aligned}
q_a{}^c q_b{}^d R_{cd} &\triangleq \delta_a^A \delta_b^B \Big\{ R_{AB}[\Omega] - (\partial_v + \kappa)\lambda_{AB} - 2D_{(A}\omega_{B)} - 2\omega_A\omega_B \\
&\quad + 2\sigma^C{}_{(A}\left[\lambda_{B)C} - \frac{1}{4}\Omega_{B)C}\lambda^D{}_D\right] - \frac{d-6}{2(d-2)}\Theta\left[\lambda_{AB} + \frac{1}{d-6}\Omega_{AB}\lambda^C{}_C\right]\Big\}.
\end{aligned} \tag{2.12}$$

In the above expressions $R_{AB}[\Omega]$ and $D_A$ are the Ricci tensor and the covariant derivative compatible with the horizon spatial metric $\Omega_{AB}$. An important remark here is that $\Omega_{AB}$ is unconstrained by the field equations, i.e. it enters the description of the horizon as free data; see Table 2.1.

The evolution of the longitudinal shear $\sigma_{AB}$, on the other hand, is independent from Einstein equations, as it involves the Weyl tensor. It is known as the 'tidal force equation' [5, 77, 91]

$$\ell^c q_a{}^d \ell^e q_b{}^f C_{cdef} \triangleq -\delta_a^A \delta_b^B \left[(\partial_v - \kappa)\sigma_{AB} - \sigma_{AC}\sigma_B{}^C - \frac{1}{d-2}\Omega_{AB}\sigma_{CD}\sigma^{CD}\right]. \tag{2.13}$$

Note that the Raychaudhuri (2.10) and tidal force (2.13) equations are part of Sachs's optical scalar equations [5, 91].

---

[7]We use the sign convention $R^a{}_{bcd} = 2\partial_{[c}\Gamma^a_{d]b} + \dots$ for the Riemann tensor.

In $d = 4$, for any 2-dimensional symmetric tensor $\sigma_{AB}$, $\sigma_{AC}\sigma_B{}^C = \frac{1}{2}\Omega_{AB}\sigma_{CD}\sigma^{CD}$ and thus the above tidal force equations agrees with the $d = 4$ expression given in Eq. (6.31) of Ref. [77]. As for the deformation rate evolution equation, it agrees with Eq. (6.43) of Ref. [77][8].

**Non-expanding and isolated horizons**   At this point it is instructive to make contact with the notion of non-expanding horizons (NEHs) introduced by Ashtekar et al. in Refs. [12, 30, 31]. A NEH is a codimension-1 submanifold of the $d$-dimensional spacetime such that:

   (a) it is a null surface of topology $\mathbb{R} \times \mathbb{S}^{d-2}$;

   (b) the expansion of every null normal $\ell^a$ vanishes on the horizon; and,

   (c) the spacetime Ricci tensor satisfies $R^a{}_b\ell^b \,\hat{\propto}\, \ell^a$.

The NEH requirement is an intrinsic property of a null hypersurface and provides a good (local) description of black holes in quasi-equilibrium[9].

   For the horizon to be a NEH, we therefore see the defining condition that its area is constant, $\partial_v\sqrt{\Omega} = 0$. Furthermore, the third condition[10] above is the null Raychaudhuri equation [86], $\ell^a\ell^b R_{ab} \hat{=} 0$, and the Damour equation [87, 88], $q_a{}^b\ell^c R_{bc} \hat{=} 0$. The null Raychaudhuri equation (2.10) implies $\sigma_{AB} \hat{=} 0$, since $\Omega_{AB}$ is positive-definite, which requires a stationary horizon spatial metric, while the Damour equation (2.11) further constraints the time dependence of the twist 1-form field to be $\partial_v\omega_A \hat{=} D_A\kappa$. The null normals to a NEH are then, besides twist-free, also shear-free and expansion-free. In particular, an *extremal* NEH has

$$\tilde{\kappa} \hat{=} 0, \quad \sigma_{AB} \hat{=} 0, \quad \Theta \hat{=} 0, \quad \partial_v\omega_A \hat{=} 0, \tag{2.14}$$

or, equivalently, in terms of near-horizon metric fields,

$$\kappa = 0, \quad \partial_v\Omega_{AB} = 0, \quad \partial_v\vartheta_A = 0. \tag{2.15}$$

A subclass of NEHs are isolated horizons (IHs) [12, 30, 31], that is, NEHs with $\partial_v\lambda_{AB} = 0$. We remark here that every Killing horizon is an IH, but the converse is not true; see for instance Refs. [31, 94].

## 2.2   Geometry near null infinity

On the other side of the proposed correspondence is another well-known null surface that is associated with asymptotically flat spacetimes: null infinity, $\mathscr{I}$. The analog of the null Gaussian coordinate system we used to describe the near-horizon geometry is the (algebraic[11]) Newman-Unti (NU) gauge [78]; the spacetime near future null infinity $\mathscr{I}^+$ is charted by a retarded time coordinate $u$, whose level-sets are null surfaces, an affine[12] radial coordinate $r$

---

[8]The necessary matchings of notations are, $\Theta_{ab}^{[77]} = \Sigma_{ab}^{\text{here}} = \sigma_{ab}^{\text{here}} - \frac{1}{d-2}q_{ab}\Theta^{\text{here}}$, $\theta^{[77]} = \Theta^{\text{here}}$, $\Omega_a^{[77]} = \omega_a^{\text{here}}$, $\kappa^{[77]} = \kappa^{\text{here}}$, $\Xi_{ab}^{[77]} = \Xi_{ab}^{\text{here}} \hat{=} \delta_a^A\delta_b^B\left[-\frac{1}{2}\lambda_{AB}^{\text{here}}\right]$ and $\theta_{(k)}^{[77]} = q^{ab}\Xi_{ab}^{[77]} \hat{=} -\frac{1}{2}\Omega_{\text{here}}^{AB}\lambda_{AB}^{\text{here}}$, the minus signs coming from the convention $\ell \cdot n = -1$ that we use here.

[9]NEHs were first studied by Hájíček under the name of "perfect horizons" in Ref. [92] and they are closely related to the notions of trapped surfaces [9], apparent horizons [93] and trapping horizons [10]; see Ref. [77] for more details.

[10]See footnote 2 of Ref. [33] for more details on the origin of this requirement.

[11]Alternatively, one can consider the differential NU gauge, with $\partial_r g_{ur} = 0$; see Ref. [95].

[12]In Bondi gauge [3, 96], the radial coordinate is instead an areal distance; see Refs. [95, 97, 98] for more details on the relation between Bondi and NU gauges.

of the null generators and $d-2$ transverse spatial coordinates $x^A$ that are parallel transported along the null generators. In this gauge, the metric and its inverse can be read from

$$ds^2_{\mathscr{I}^+} = -F\,du^2 - 2\,du\,dr + r^2 \mathcal{H}_{AB}\left(dx^A - \frac{U^A}{r^2}du\right)\left(dx^B - \frac{U^B}{r^2}du\right),$$

$$\partial^2_{\mathscr{I}^+} = F\partial^2_r - 2\,\partial_u\partial_r - 2\,\frac{U^A}{r^2}\partial_r\partial_A + \frac{1}{r^2}\mathcal{H}^{AB}\partial_A\partial_B\,,$$

(2.16)

with $\mathcal{H}^{AB}$ the components of the inverse of the spatial metric $\mathcal{H}_{AB}$. Asymptotic flatness requires the following boundary conditions for the asymptotic metric fields[13]

$$\mathcal{H}_{AB}\left(u,r,x^C\right) = q_{AB}\left(x^C\right) + \frac{1}{r}C_{AB}\left(u,x^C\right) + o\left(r^{-1}\right),$$

$$F\left(u,r,x^A\right) = \frac{R[q]}{(d-2)(d-3)} + \frac{1}{d-2}\partial_u C_A{}^A - \frac{2m_{\mathrm{B}}}{r} + o\left(r^{-1}\right),$$

$$U^A\left(u,r,x^B\right) = \frac{1}{2(d-3)}\left(D_B C^{AB} - D^A C_B{}^B\right) + \frac{2}{3r}\left[N^A - \frac{1}{2}C^{AB}D^C C_{BC}\right] + o\left(r^{-1}\right).$$

(2.17)

In the above, $D_A$ and $R[q]$ denote the covariant derivative and curvature associated with the boundary metric $q_{AB}$, and $C_{AB}$ is an arbitrary function of $(u,x^A)$. We remark here that the boundary spatial metric $q_{AB}$ is taken to be fixed on $\mathscr{I}$, while $\partial_u q_{AB} = 0$ by virtue of the leading order field equations. In four spacetime dimensions, the subleading asymptotic fields $m_{\mathrm{B}}\left(u,x^A\right)$ and $N_A\left(u,x^B\right)$ are the Bondi mass and angular momentum aspects respectively; they enter as integration constants whose time evolution is constrained by the field equations [3,5]. In the same spirit, the STF parts of the subleading asymptotic fields in the spatial metric also enter as data; the evolution of $C_{\langle AB\rangle}$ is completely unconstrained[14], that is, it comprises the free data, while the dynamics of the successive $o\left(r^{-1}\right)$ fields are fixed by the field equations. In particular, $C_{\langle AB\rangle}$ is the asymptotic gravitational shear tensor and encodes the polarization modes of the gravitational waves. For instance, the gravitational wave energy flux through $\mathscr{I}^+$ is captured by the (square of the) Bondi news tensor[15] $N_{AB} = \partial_u C_{\langle AB\rangle} - 2\omega^{-1}D_{\langle A}D_{B\rangle}\omega$, $\omega$ here being the conformal factor that relates the boundary metric to the spherical metric, $q_{AB} = \omega^2 \gamma_{AB}$. The subleading shear tensors can be related to multipole moments [40].

As it was observed in Refs. [31–34], null infinity can be incorporated within the framework of NEHs; namely, $\mathscr{I}^+$ is a weakly isolated horizon for the conformal spacetime,

$$d\tilde{s}^2_{\mathscr{I}^+} = \Omega^2 ds^2_{\mathscr{I}^+}, \quad \Omega^2 = \frac{\alpha^2}{r^2},$$

(2.18)

where $\alpha$ is some length scale that we leave implicit at the current stage. To see this more explicitly, one resorts to the definition of asymptotic flatness near null infinity [3, 5, 80, 96, 106, 110–113]. For concreteness, we take Definition 1 in Ref. [80] and denote $g_{ab}$ the physical metric (which solves Einstein equations $R_{ab} - \frac{1}{2}g_{ab}R = 8\pi T_{ab}$), $\tilde{g}_{ab} = \Omega^2 g_{ab}$ (with $\Omega$ a smooth function such that $\Omega \doteq 0$) the unphysical metric and $n_a := \tilde{\nabla}_a\Omega$ is nowhere vanishing on $\mathscr{I}$. From the field equations for $\tilde{g}_{ab}$,

$$\tilde{R}_{ab} - \frac{1}{2}\tilde{g}_{ab}\tilde{R} + (d-2)\Omega^{-1}\left[\tilde{\nabla}_a n_b - \tilde{g}_{ab}\tilde{\nabla}_c n^c\right] + \frac{(d-1)(d-2)}{2}\Omega^{-2}\tilde{g}_{ab}n_c n^c = 8\pi\Omega^2\hat{T}_{ab}\,,$$

(2.19)

---

[13]See e.g. Refs. [95,99–105] for relaxations of these boundary conditions.

[14]The trace $C_A{}^A = q^{AB}C_{AB}$ controls the origin for the affine parameter of null generators and can, in particular, be freely set to zero in the NU gauge [98].

[15]The second term in the news tensor, besides $\partial_u C_{\langle AB\rangle}$, is required when the boundary metric is not spherical. It follows from the Geroch tensor [106], and ensures that the news tensor is, besides traceless, also independent of the choice of the conformal completion [80,107], see also Refs. [108,109].

292  with $\hat{T}_{ab} := \Omega^{-2}T_{ab}$ admitting a smooth limit to $\mathscr{I}$, one can extract the following implica-
293  tions [15, 33, 110]:

(a)  $n_a n^a \triangleq 0$, i.e. $\mathscr{I}$ is a null hypersurface, and $n^a$ is a null normal.

(b)  The gauge freedom can be used to go to divergence-free conformal frames, for which
     $\tilde{\nabla}_a n^a \triangleq 0$. As such the expansion of all normals vanishes at $\mathscr{I}$. Equations (2.19) then
     further imply $\tilde{\nabla}_a n^b \triangleq 0$ all together, and, hence, the twist 1-form on $\mathscr{I}$ vanishes as well.

(c)  The Schouten tensor $\tilde{S}_{ab} = \tilde{R}_{ab} - \frac{1}{2(d-1)}\tilde{g}_{ab}\tilde{R}$ of the unphysical metric $\tilde{g}_{ab}$ satisfies $\tilde{S}^a{}_b n^b \triangleq -f n^a$,
     with $f = \frac{d-2}{2}\Omega^{-2}n_c n^c$. Therefore, $\tilde{R}^a{}_b n^b \triangleq \zeta n^a$, with $\zeta \triangleq \frac{\tilde{R}}{2(d-1)} - f$.

(d)  If $\tilde{g}_{ab}$ is $C^3$, the Weyl tensor $\tilde{C}^a{}_{bcd}$ vanishes on $\mathscr{I}$. Hence, $\Omega^{-1}\tilde{C}_{abcd}$ admits a continuous
     limit to $\mathscr{I}$.

302  We see, therefore, that the fall-offs $T_{ab} = \mathcal{O}(\Omega^2)$ ensure that the unphysical conformally
303  completed spacetime $(\tilde{\mathcal{M}}, \tilde{g}_{ab})$ contains a NEH at the boundary, i.e. at null infinity, even in
304  the presence of radiation [32, 33]. More explicitly, null infinity is a codimension-1 null surface
305  of topology $\mathbb{R} \times \mathbb{S}^{d-2}$ by definition, all null normals are expansion-free there and the null
306  Raychaudhuri and Damour equations in the unphysical spacetime are trivially satisfied,

$$\tilde{R}_{uu} \triangleq 0, \quad \tilde{R}_{uA} \triangleq 0 \quad \tilde{C}_{uAuB} \triangleq 0, \tag{2.20}$$

307  while the last (tidal force) equation is just the statement that $\Psi_0 \triangleq 0$ on a NEH [30, 31, 114,
308  115]. Furthermore, the vanishing of $\tilde{\nabla}_a n^b$ on $\mathscr{I}$ means that this NEH is non-rotating and ex-
309  tremal, properties which arise (as the expansion-free condition) from the existence of preferred
310  divergence-free conformal frames [32, 33].

## 2.3  Null infinity as a spatially inverted extremal horizon

312  From what we just discussed, it follows that a conformally completed spacetime whose bound-
313  ary is $\mathscr{I}$ (as defined above) is diffeomorphic to a geometry that contains an extremal horizon
314  at a finite distance, as also pointed out in Ref. [71]. This can be seen explicitly by performing
315  the following spatial inversion

$$r = \frac{\alpha^2}{\rho}, \quad u = v, \tag{2.21}$$

316  where $\alpha$ is an arbitrary constant length scale introduced in Eq. (2.18), which maps the con-
317  formal geometry to the one around a horizon upon identifying (see Eqs. (2.1) and (2.16))

$$\begin{aligned}
d\tilde{s}^2_{\mathscr{I}^+} &= ds^2_{\mathscr{H}^+} \quad \text{with} \\
\mathcal{F}(v, \rho, x^A) &= \alpha^{-2}F\left(u \mapsto v, r \mapsto \frac{\alpha^2}{\rho}, x^A\right), \\
g_{AB}(v, \rho, x^C) &= \alpha^2 \mathcal{H}_{AB}\left(u \mapsto v, r \mapsto \frac{\alpha^2}{\rho}, x^C\right) \quad \text{and} \\
\theta^A(v, \rho, x^B) &= -\rho\alpha^{-4}U^A\left(u \mapsto v, r \mapsto \frac{\alpha^2}{\rho}, x^B\right).
\end{aligned} \tag{2.22}$$

318  The spatial inversion exactly maps an $r \to \infty$ (near-$\mathscr{I}^+$) asymptotic expansion to a near-
319  horizon $\rho \to 0$ expansion (see Fig. 1). In order for the fall-off conditions to be preserved,
320  we see then that such an interpretation of null infinity as a finite-distance horizon requires
321  the latter to be *extremal*, $\kappa = 0$, and *non-rotating*, $\omega_A \triangleq 0$. The explicit dictionary as well as
322  the interpretation and dynamics of the various quantities from the two sides is displayed in

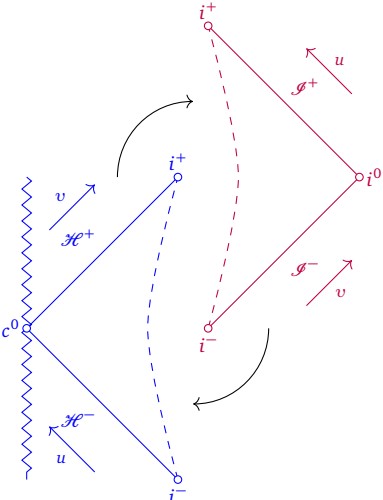

Figure 1: Penrose diagram representation of the conformal isomorphism between an asymptotically flat spacetime and the geometry near a dual extremal horizon. The spatial inversion in Eq. (2.21) (black arrows) conformally maps the geometry near $\mathscr{I}^+$ of an asymptotically flat spacetime (red partial Penrose diagram) to that near a future horizon $\mathscr{H}^+$ that is extremal, non-expanding and non-rotating (blue partial Penrose diagram), and vice versa. An exactly analogous spatial inversion maps the geometry near $\mathscr{I}^-$ to that near a past horizon $\mathscr{H}^-$ with the same properties.

Table 2.1. In particular, the free data at $\mathscr{I}$ (the asymptotic shear $C_{AB}$) is mapped to the horizon transversal shear, while the horizon free data $\Omega_{AB}$ corresponds to the sphere metric $q_{AB}$, which is fixed at $\mathscr{I}$.

| $\mathscr{H}$ | Name | Evolution equation | $\mathscr{I}$ |
|---|---|---|---|
| $\kappa$ | surface gravity | | 0 |
| $\Theta$ | expansion | null Raychaudhuri (2.10) | 0 |
| $\Omega_{AB}$ | horizon metric | (free data) | $q_{AB}$ (fixed) |
| $\omega_A$ | (Hájíček) twist | Damour (2.11) | 0 |
| $\sigma_{AB}$ | longitudinal shear | derived from (2.8) | 0 |
| $\lambda_{AB}$ | transversal shear | transversal deformation rate evolution (2.12) | $C_{AB}$ (free data) |

Table 2.1: Summary of the dictionary between the quantities appearing in the near-horizon geometry (2.1) and an asymptotically flat spacetime (2.16) that are conformally mapped onto each other under the spatial inversion of Eq. (2.21).

## Physics at the two boundaries

It is important to recall that the null infinity-side of this 'duality' refers to the conformal completion of an asymptotically flat spacetime, in contrast to the horizon-side of the correspondence which can reside in the physical spacetime. This results in very different physics at the two null boundaries, as already emphasized in Refs. [32, 33]. In particular, the physics at null infinity generically involves the presence of radiation, without ruining the interpretation of $\mathscr{I}$ as a weakly isolated horizon in the conformally completed spacetime.

A direct consequence of the fact that $\mathscr{I}$ is a NEH in the unphysical spacetime even in the presence of radiation is that there is a non-trivial energy-momentum tensor induced at the 'dual' horizon. To see this, focusing to $d = 4$ spacetime dimensions, recall first the dictionary mapping the near-horizon spatial metric $\Omega_{AB}$ and transversal shear $\lambda_{AB}$ to the $\mathscr{I}$ spatial metric

$q_{AB}$ and asymptotic shear $C_{AB}$,

$$\Omega_{AB} = \alpha^2 q_{AB}, \quad \lambda_{AB} = C_{AB}. \tag{2.23}$$

As mentioned above, from the horizon-side, $\Omega_{AB}$ is free data and $\partial_v \lambda_{AB}$ is constrained by the field equations, while, from the null infinity-side, $q_{AB}$ is a universal structure fixed at $\mathscr{I}$ (typically taken to be the unit round metric on $\mathbb{S}^2$), and the Bondi news tensor $N_{AB} = \partial_u C_{\langle AB \rangle}$ that encodes free data. From the horizon geometry point of view, the Einstein field equations

$$\tilde{R}_{\langle AB \rangle} = 8\pi \tilde{T}_{\langle AB \rangle}, \tag{2.24}$$

provide the effective energy-momentum tensor associated with radiation in the physical space-time near null infinity

$$8\pi \tilde{T}_{\langle AB \rangle} \hateq N_{AB}. \tag{2.25}$$

As a last remark, let us note that the map described above is true for any asymptotically flat spacetime, regardless of whether the latter contains a genuine horizon at finite distance or not. Even if the spacetime does contain a horizon, the latter has in general nothing to do with the extremal horizon dual to null infinity. A very special exception is the four-dimensional extremal Reissner-Nordström black hole, i.e. an asymptotically flat, static and spherically symmetric geometry that solves the general-relativistic electrovacuum field equations and contains an extremal event horizon. This has the peculiarity of being equipped with a discrete conformal isometry of the form just described, first pointed out by Couch & Torrence [76]. The Couch-Torrence isometry can thus be understood as a consequence of the general geometric $\mathscr{H}/\mathscr{I}$ duality described above. We will discuss in the next section some of its direct physical implications. In general, spacetimes that are 'self-dual' under the spatial inversions described in this section are by definition extremal black hole geometries, since for these cases one automatically has information about the global structure of the geometry that contains the extremal event horizon.

## 2.4 Near-horizon vs asymptotic symmetries

We end this section by reviewing and contrasting the near-horizon symmetry analysis with the one near null infinity. For another comparison of the symmetry groups with different boundary conditions at $\mathscr{I}$ and horizons, see Ref. [116]. The set of symmetries preserving a certain notion of asymptotic flatness as the metric approaches null infinity has long been known to span the BMS group (see e.g. Ref. [117] for a recent review). Understanding the nature of analogous symmetries near (non-extremal) black hole horizons is, to a large extent, a much more recent enterprise. For generic horizons, the near-horizon symmetries were first analyzed in Refs. [44, 45][16], where they were showed to span a bigger set than the BMS symmetries, as the supertranslation parameter is allowed to be an arbitrary function of advanced time as well, spanning the so-called Newman-Unti algebra (see e.g. Ref. [124]). Of course, as already emphasized, this difference can be traced back to the fact that finite-distance horizons are null sub-regions of the physical spacetime, rather than the conformally completed spacetime.

Given the intrinsic Carrollian nature of these two null hypersurfaces [19, 22–24, 89, 106], their symmetry-preserving structure shares several similarities[17] but also important differences. After a brief review in terms of the unified framework of Carrollian symmetries, the presentation below aims to unify the treatment of asymptotic symmetries for both $\mathscr{I}$ and $\mathscr{H}$ by treating the gauge-fixing and respective boundary conditions successively.

---

[16]See Refs. [36, 46–48, 50, 52–54, 115, 118–123] for further works.

[17]We do not discuss here potential matching of their respective asymptotic symmetry parameters; see Refs. [47, 75, 125–127] for works in this direction.

### Extended Carroll symmetries

Let $C = \Sigma^{d-2} \times \mathbb{R}$ be a $(d-1)$-dimensional smooth Carroll manifold ($\Sigma$ denotes a Riemannian manifold), endowed with a metric g whose kernel is generated by a nowhere vanishing vector field n [81].

The conformal Carroll algebra of level $N$, $\mathfrak{ccarr}_N$, is spanned by vector fields $\xi$ such that

$$\mathcal{L}_\xi \mathrm{g} = \lambda \mathrm{g} \, , \qquad \mathcal{L}_\xi \mathrm{n} = -\frac{\lambda}{N} \mathrm{n}, \tag{2.26}$$

for some function $\lambda$ and positive integer $N$ [81]. Introducing coordinates $(u, x^A)$ on $C$ such that $\mathrm{n} = \partial_u$ and $\mathrm{g} = g_{AB} dx^A dx^B$, the generic expression for such vector fields is

$$\xi = Y^A(x)\partial_A + \left( T(x) + u\frac{\lambda}{N} \right)\partial_u \, , \qquad \text{where } \lambda = \frac{2}{d-2}D_A Y^A, \tag{2.27}$$

with $Y^A$ a conformal Killing vector field of $\Sigma^{d-2}$ and $T$ is a density of conformal weight $-2/N$. For $\Sigma^{d-2} = \mathbb{S}^{d-2}$ endowed with its round metric, we thus immediately see that the conformal Carroll transformations of level $N = 2$ are the semi-direct product of the conformal group of $\mathbb{S}^{d-2}$ together with supertranslations $T$ (of conformal weight $-1$), hence the celebrated isomorphism $\mathfrak{bms}^d = \mathfrak{ccarr}_2^{d-1}$ [81].

The Newman-Unti Lie algebra [17, 78, 81, 124], $\mathfrak{nu}$ is more generic as it does not require preserving the strong conformal geometry. It is spanned by all vector fields $\zeta$ on $C$ such that

$$\mathcal{L}_\zeta \mathrm{g} = \lambda \mathrm{g}. \tag{2.28}$$

This condition automatically implies that the direction of n is preserved. In Carrollian coordinates $(u, x^A)$, the $\mathfrak{nu}$ vector fields are[18]

$$\zeta = Y^A(x)\partial_A + f(u, x)\partial_u \, , \tag{2.29}$$

with $Y^A$ a conformal Killing vector field of $\Sigma^{d-2}$ and $f$ is now an arbitrary function of $x^A$ and $u$. As opposed to BMS supertranslations, the functions $f$ do not form an abelian ideal. As we recall below, the $\mathfrak{nu}$ algebra is preserving the Carrollian structure of a generic horizon [24, 45]. An interesting subalgebra of the Newman-Unti algebra was highlighted in Ref. [17] as the algebra defined by

$$\mathcal{L}_\zeta \mathrm{g} = \lambda \mathrm{g} \, , \qquad (\mathcal{L}_\mathrm{n})^n \xi = 0. \tag{2.30}$$

This subalgebra, denoted $\mathfrak{nu}_n$, is spanned by vector fields of the form of Eq. (2.29) with the restriction

$$\partial_u^n f = 0. \tag{2.31}$$

We will see below that the near-horizon symmetries of an extremal horizon span the $\mathfrak{nu}_2$ algebra [45, 52]. Notice also the relationship $\mathfrak{nu}_1 = \mathfrak{ccarr}_\infty$ [17].

### Newman-Unti gauge

Near a smooth null hypersurface located at r = 0, one can always choose null Gaussian coordinates v, r, $x^A$ in which the metric satisfies $g_{rr} = g_{rA} = 0$, $g_{vr} = 1$ [79]. Both the near-horizon geometry and null infinity can be written in the Newman-Unti (NU) form, as done in Eq. (2.1) and Eq. (2.16). Independently of the location of the null hypersurface (be it at a finite or infinite distance in spacetime), one can thus first search for the generic form of vector fields preserving the NU gauge. The conditions

$$\mathcal{L}_\zeta g_{ra} = 0 \, , \tag{2.32}$$

---

[18]They generate what were called Carrollian diffeomorphisms in Ref. [90].

can be seen to lead to the generic form

$$
\begin{aligned}
\zeta^{\text{v}} &= f \,, \\
\zeta^{\text{r}} &= -\text{r}\, \partial_{\text{v}} f + Z + J \,, \\
\zeta^{\text{A}} &= Y^{\text{A}} + I^{\text{A}} \,,
\end{aligned}
\tag{2.33}
$$

where NU supertranslations $f$, superrotations $Y^A$ and radial transformations $Z$ satisfy $\partial_{\text{r}} f = 0 = \partial_{\text{r}} Y^A = \partial_{\text{r}} Z$ but can depend arbitrarily of $(\text{v}, \text{x}^{\text{A}})$ at this stage. The remaining functions satisfy $\partial_{\text{r}} J = g^{\text{AB}} g_{\text{vA}} \partial_{\text{B}} f$, $\partial_{\text{r}} I^A = g^{\text{AB}} \partial_{\text{B}} f$.

Near $\mathscr{I}^+$, this gives (adapting to retarded time) the following asymptotic Killing vector field [98]

$$
\begin{aligned}
\xi^u &= f \,, \\
\xi^r &= -r\, \partial_u f + Z + J \,, \qquad J = -\partial_A f \int_r^{\infty} dr'\, g^{r\text{A}} \,, \\
\xi^A &= Y^A + I^A \,, \qquad I^A = -\partial_B f \int_r^{\infty} dr'\, g^{AB} \,.
\end{aligned}
\tag{2.34}
$$

In near-horizon (advanced) coordinates, the corresponding asymptotic Killing vector field near $\mathscr{H}^+$ reads [44, 45]

$$
\begin{aligned}
\chi^{\upsilon} &= f \,, \\
\chi^{\rho} &= \mathcal{Z} - \rho\, \partial_{\upsilon} f + \mathcal{J} \,, \qquad \mathcal{J} = \partial_A f \int_0^{\rho} d\rho'\, g^{AB} g_{\upsilon B} \,, \\
\chi^A &= \mathcal{Y}^A + \mathcal{I}^A \,, \qquad \mathcal{I}^A = -\partial_B f \int_0^{\rho} d\rho'\, g^{AB} \,.
\end{aligned}
\tag{2.35}
$$

**Asymptotic symmetries near $\mathscr{I}^+$**

On top of the gauge preserving conditions solved above, the asymptotic Killing vectors are also subject to boundary conditions. Near $\mathscr{I}^+$, asymptotic flatness imposes [3, 5, 96]

$$
\mathcal{L}_\xi g_{uA} = \mathcal{O}\left(r^0\right) \,, \qquad \mathcal{L}_\xi g_{AB} = \mathcal{O}(r) \,, \qquad \mathcal{L}_\xi g_{uu} = \mathcal{O}\left(r^{-1}\right) \,,
\tag{2.36}
$$

leading to strong restrictions on the asymptotic Killing vector of Eq. (2.34). The first condition is

$$
\mathcal{L}_\xi g_{uA} = \mathcal{O}\left(r^0\right) \quad \Rightarrow \partial_u Y^A = 0 \,.
\tag{2.37}
$$

The second condition of fixed boundary metric on the celestial sphere imposes that the superrotations, $Y^A$, are constrained to be conformal Killing vectors of $q_{AB}$,

$$
\mathcal{L}_\xi g_{AB} = \mathcal{O}(r) \Rightarrow \mathcal{L}_Y q_{AB} = \frac{2}{d-2} q_{AB} D_C Y^C \,, \quad \partial_u f = \frac{1}{d-2} D_A Y^A \,.
\tag{2.38}
$$

We can thus write

$$
f = T(x^A) + u X(x^A) \,, \qquad X = \frac{1}{d-2} D_C Y^C \,.
\tag{2.39}
$$

The last boundary condition of Eq. (2.36) does not impose further constraints.

The residual symmetry parameter $Z$ in the radial component of the asymptotic Killing vector is associated with the choice of origin for the affine parameter of the null geodesic [98]. This residual freedom can be used to set to zero the trace $C_A{}^A = 0$, which fixes

$$
Z\left(u, x^A\right) = -\frac{1}{d-2} D^2 f \,.
\tag{2.40}
$$

In Bondi gauge, the trace condition $C_A^A = 0$ is instead implemented by the determinant condition. The authors of Ref. [95] have argued that the NU gauge is thus in a sense 'less restrictive' than the Bondi gauge, as it generally allows for an arbitrary radial translation $Z$[19]. Putting everything together, one thus gets

$$\xi = T\left(x^A\right)\partial_u + Y^A\left(x^B\right)\partial_A + X\left(x^A\right)\left(u\,\partial_u - r\,\partial_r\right)$$
$$- \frac{1}{d-2}\left(D^2 T + uD^2 X\right)\partial_r + \frac{1}{r}\left(D_B T + uD_B X\right)\left(\mathcal{U}^B \partial_r - \mathcal{H}^{AB}\partial_A\right), \tag{2.41}$$

where $X = \frac{1}{d-2}D_A Y^A$ and we have defined

$$\frac{1}{r}\mathcal{U}^A\left(u, r, x^B\right) := \int_r^\infty \frac{dr'}{r'^2}U^A\left(u, r', x^B\right),$$
$$\frac{1}{r}\mathcal{H}^{AB}\left(u, r, x^C\right) := \int_r^\infty \frac{dr'}{r'^2}\mathcal{H}^{AB}\left(u, r', x^C\right), \tag{2.42}$$

such that $\mathcal{U}^A\left(u, r, x^B\right) = U^A\left(u, r, x^B\right) + o\left(r^{0^-}\right)$ and $\mathcal{H}^{AB}\left(u, r, x^C\right) = q^{AB}\left(x^C\right) + o\left(r^{0^-}\right)$.

The above vector fields preserve the entire leading-order structure of the metric near $\mathscr{I}^+$, while their action on the traceless gravitational shear is

$$\delta_\xi C_{AB} = \left(f\,\partial_u + \mathcal{L}_Y - \frac{1}{d-2}D_C Y^C\right)C_{AB} - 2D_{\langle A}D_{B\rangle}f. \tag{2.43}$$

**Asymptotic symmetries near $\mathscr{H}^+$**

Let us first briefly comment on the role of the radial translation $\mathcal{Z}$ in the horizon Killing vector field of Eq. (2.35). This captures angle-dependent shifts of the location of the horizon, $\rho = 0 \rightarrow \rho = \mathcal{Z}$. As such, choosing to preserve the horizon location at the origin of the affine parameter $\rho$, we set it to $\mathcal{Z} = 0$ (as in Ref. [45]).

Now, the near-horizon boundary conditions are [45]

$$\mathcal{L}_\chi g_{vA} = \mathcal{O}(\rho), \quad \mathcal{L}_\chi g_{AB} = \mathcal{O}(\rho^0), \quad \mathcal{L}_\chi g_{vv} = \begin{cases} \mathcal{O}(\rho) & \text{for } \kappa \neq 0 \\ \mathcal{O}(\rho^2) & \text{for } \kappa = 0 \end{cases}. \tag{2.44}$$

The first condition imposes time-independence of the superrotation parameters

$$\mathcal{L}_\chi g_{vA} = \mathcal{O}(\rho) \quad \Rightarrow \quad \partial_v \mathcal{Y}^A = 0. \tag{2.45}$$

For the generic non-extremal case ($\kappa \neq 0$), the rest of the boundary conditions do not lead to further constraints on the form of the vector field, and we get the nu vector fields of Eq. (2.29). However, for an extremal horizon ($\kappa = 0$), we get the constraint

$$\mathcal{L}_\chi g_{vv}^{\text{ext}} = \mathcal{O}\left(\rho^2\right) \quad \Rightarrow \partial_v^2 f_{\text{ext}} = 0 \quad \Rightarrow f_{\text{ext}} = \mathcal{T}(x^A) + v\mathcal{X}(x^A). \tag{2.46}$$

For an extremal horizon, we thus get

$$\chi^{\text{ext}} = \mathcal{T}\left(x^A\right)\partial_v + \mathcal{X}\left(x^A\right)\left(v\,\partial_v - \rho\,\partial_\rho\right) + \mathcal{Y}^A\left(x^B\right)\partial_A$$
$$+ \rho\left(D_B \mathcal{T} + vD_B \mathcal{X}\right)\left(\frac{1}{2}\Theta^B \rho\,\partial_\rho - \mathcal{G}^{AB}\partial_A\right). \tag{2.47}$$

---

[19]Notice, however, that since this transformation is subleading in $r$, it does not affect the Carrollian structure.

The supertranslations, $\mathcal{T}$, superdilatations, $\mathcal{X}$, and superotations, $\mathcal{Y}^A$, all live on the horizon spatial cross-sections, and we have defined

$$
\begin{aligned}
\frac{1}{2}\rho^2\Theta^A\left(v,\rho,x^B\right) &:= \int_0^\rho d\rho'\,\rho'\,\theta^A\left(v,\rho',x^B\right),\\
\rho\,\mathcal{G}^{AB}\left(v,\rho,x^C\right) &:= \int_0^\rho d\rho'\,g^{AB}\left(v,\rho',x^C\right),
\end{aligned}
\tag{2.48}
$$

such that $\Theta^A\left(v,\rho,x^B\right) = \vartheta^A\left(v,x^B\right)+o\left(\rho^{0^+}\right)$ and $\mathcal{G}^{AB}\left(v,\rho,x^C\right) = \Omega^{AB}\left(v,x^C\right)+o\left(\rho^{0^+}\right)$. Their action on the leading-order asymptotic fields can then be worked out to be

$$
\begin{aligned}
\delta_\chi\Omega_{AB} &= f\,\partial_v\Omega_{AB}+\mathcal{L}_\mathcal{Y}\Omega_{AB},\\
\delta_\chi\vartheta^A &= f\,\partial_v\vartheta^A+\mathcal{L}_\mathcal{Y}\vartheta^A-2\Omega^{AB}D_B\mathcal{X}-\partial_v\Omega^{AB}D_Bf,\\
\delta_\chi\mathcal{F}_0 &= f\,\partial_v\mathcal{F}_0+\mathcal{L}_\mathcal{Y}\mathcal{F}_0-3\vartheta^AD_A\mathcal{X}-\partial_v\vartheta^AD_Af.
\end{aligned}
\tag{2.49}
$$

For an extremal NEH, for which one additionally has $\partial_v\Omega_{AB}=0$ and $\partial_v\vartheta^A=0$, these reduce to

$$
\begin{aligned}
\delta_\chi\Omega_{AB} &= \mathcal{L}_\mathcal{Y}\Omega_{AB},\\
\delta_\chi\vartheta^A &= \mathcal{L}_\mathcal{Y}\vartheta^A-2\Omega^{AB}D_B\mathcal{X},\\
\delta_\chi\mathcal{F}_0 &= f\,\partial_v\mathcal{F}_0+\mathcal{L}_\mathcal{Y}\mathcal{F}_0-3\vartheta^AD_A\mathcal{X}.
\end{aligned}
\tag{2.50}
$$

The action of the above near-$\mathcal{H}^+$ asymptotic Killing vectors, in particular, preserves the character of the extremal NEH without any further constraints on $\chi$. If now one deals with a non-rotating extremal NEH, i.e. with $\vartheta^A=0$, then the superdilatations are reduced to global rescalings, namely $\mathcal{X}\left(x^A\right)=$ const.

# 3 A self-inverted example: The case of extremal Reissner-Nordström black hole

In this section, we consider a known example of extremal black hole geometry which has the property of being 'self-dual' under the spatial inversion discussed in the previous section. This is the four-dimensional extremal Reissner-Nordström (ERN) black hole for which the spatial inversion of Section 2 becomes the discrete Couch-Torrence conformal isometry identified in [76]. Utilizing this property, it is possible to extract new pairings between near-horizon and near–null infinity data which dictate the one-to-one matching between infinite towers of conserved quantities. Previous literature [61,68–71,74,75] has focused on the case of a probe scalar and Maxwell field in the ERN background. In this section we will extend these results to the case of gravitational and spin-weight $s$ perturbations. The treatment of gravitational perturbations require extra care compared to the scalar and spin-one cases, as we will see.

## 3.1 The symmetry

The ERN black hole geometry in four spacetime dimensions is described in Schwarzschild-like $\left(t,r,x^A\right)$ coordinates by the line element

$$
ds^2_{\text{ERN}} = -\frac{\Delta(r)}{r^2}dt^2+\frac{r^2}{\Delta(r)}dr^2+r^2d\Omega_2^2
\tag{3.1}
$$

with $d\Omega_2^2 = \gamma_{AB}\left(x^C\right)dx^Adx^B = d\theta^2+\sin^2\theta\,d\phi^2$ the line element on $\mathbb{S}^2$. The discriminant function is a perfect square, $\Delta(r) = (r-M)^2$, whose double root at $r=M$ determines the

radial location of the degenerate horizon, with $M$ the ADM mass of the black hole. This geometry describes an isolated, asymptotically flat, non-rotating and electrically charged black hole solution of the general-relativistic electrovacuum field equations, whose electric charge $Q$ attains its critical (extremal) value, $Q^2 = M^2$ (in CGS units). Besides the spherical and time translation isometries, it has a discrete conformal symmetry: the Couch-Torrence (CT) spatial inversion symmetry [76]

$$r \xrightarrow{\text{CT}} \tilde{r} = \frac{Mr}{r-M} \Rightarrow ds_{\text{ERN}}^2 = \Omega^{-2} d\tilde{s}_{\text{ERN}}^2, \quad \Omega = \frac{\tilde{r}-M}{M} = \frac{M}{r-M}, \tag{3.2}$$

where $d\tilde{s}_{\text{ERN}}^2 = -\frac{\Delta(\tilde{r})}{\tilde{r}^2} dt^2 + \frac{\tilde{r}^2}{\Delta(\tilde{r})} d\tilde{r}^2 + \tilde{r}^2 d\Omega_2^2$ is the same ERN black hole geometry, but with $\tilde{r}$ replacing $r$. In fact, as more recently noted in Ref. [128], this conformal symmetry can be realized as an *isometry* of the conformal metric, namely, $r^2 ds_{\text{ERN}}^2 = \tilde{r}^2 d\tilde{s}_{\text{ERN}}^2$. At the level of the tortoise coordinate[20]

$$r_* = r - M - \frac{M^2}{r-M} + 2M \ln \left| \frac{r-M}{M} \right|, \tag{3.3}$$

the CT inversion acts as a reflection,

$$r_* \xrightarrow{\text{CT}} -r_*, \tag{3.4}$$

that preserves the photon sphere at $r = 2M$[21] Therefore, the CT inversion maps the near-horizon region onto null infinity and vice-versa. More specifically, the future (past) event-horizon $\mathscr{H}^+$ ($\mathscr{H}^-$), specified by the $v = \text{const}$ ($u = \text{const}$) null hypersurface at $r = M$, where $v = t + r_*$ ($u = t - r_*$) is the advanced (retarded) null coordinate, gets mapped onto the future (past) null infinity $\mathscr{I}^+$ ($\mathscr{I}^-$), specified by the $u = \text{const}$ ($v = \text{const}$) null hypersurface as $r \to \infty$,

$$\mathscr{H}^\pm \overset{\text{CT}}{\longleftrightarrow} \mathscr{I}^\pm, \tag{3.5}$$

since

$$\left( v, r, x^A \right) \overset{\text{CT}}{\longleftrightarrow} \left( u, \frac{Mr}{r-M}, x^A \right). \tag{3.6}$$

The CT inversion is precisely of the form of the conformal inversion we described in Section 2. However, in contrast to the generic $\mathscr{I}/\mathscr{H}$ duality we presented there, the extremal Reissner-Nordström black hole has the characteristic property of being 'self-dual' under this conformal inversion, meaning that both $\mathscr{I}$ and the extremal horizon it describes under inversion live in the same spacetime (see Fig. 2)[22].

As already noted in Refs. [61, 75] (see also Ref. [128] for a related work), this gives the CT inversion a very physical manifestation in terms of conserved quantities: it implies that the near-$\mathscr{H}$ Aretakis charges associated with extremal black holes [57–59] are identical to the near-$\mathscr{I}$ Newman-Penrose conserved quantities [61, 65–67, 133]. We will review this matching of near-$\mathscr{H}$ and near-$\mathscr{I}$ charges explicitly for scalar [61, 68–71, 74] and electromagnetic [75] perturbations on the ERN black hole in a unified framework, and extend those results to the case of gravitational perturbations (and in fact any spin-weight $s$ perturbation).

---

[20]The integration constant has been fixed such that $r_* = 0$ corresponds to the photon sphere $r = 2M$.

[21]This geodesics point of view of the CT inversions, keeping fixed the unstable photon sphere at $r = 2M$, provides a guide for potential generalizations of these types of discrete conformal symmetries [129], and have also been utilized in Refs. [130, 131] to study physical implications on geodesic observables.

[22]See Ref. [132] for an analysis of the conformal structure of ERN spacetime.

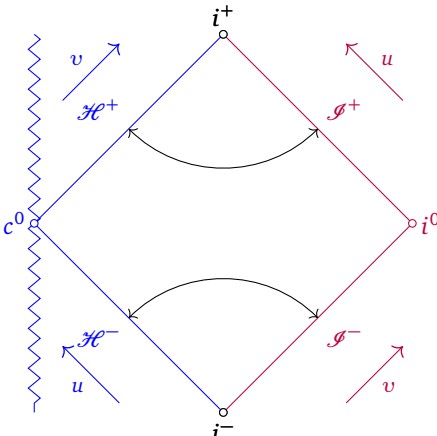

Figure 2: Part of the Penrose diagram of an extremal Reissner-Nordström black hole that describes the causally connected patch in the exterior geometry. As opposed to what happens with a generic pair of conformally related asymptotically flat spacetime and a dual geometry near an extremal horizon (see Fig. 1), the existence of the Couch-Torrence inversion (black arrows) for this geometry can be understood as the manifestation that the two null surfaces reside in the same spacetime.

## 3.2   Equations of motion for perturbations

In this section, we will deal with perturbations of the extremal Reissner-Nordström black hole and study the implications of this self-inverted conformal mapping. Naturally, we will treat the spin-weight $s$ perturbations by means of the Newman-Penrose (NP) formalism [134,135]; see Appendix A for a review of the basic elements needed and for our sign conventions.

To make contact with the notation of the previous section, let us now write down the background solution of the extremal Reissner-Nordström black hole in null Gaussian coordinates centered around the null surface of interest. To study the near-$\mathscr{I}^+$ or near-$\mathscr{I}^-$ modes, we will use retarded or advanced Eddington-Finkelstein coordinates, $\left(u, r, x^A\right)$ or $\left(v, r, x^A\right)$, respectively,

$$
\begin{aligned}
ds^2_{\text{ERN}} &= -\left(1 - \frac{M}{r}\right)^2 du^2 - 2\,du dr + r^2 d\Omega_2^2 \\
&= -\left(1 - \frac{M}{r}\right)^2 dv^2 + 2\,dv dr + r^2 d\Omega_2^2 .
\end{aligned}
\tag{3.7}
$$

A set of null tetrad vectors[23] $\{\ell, n, m, \bar{m}\}$, adapted to $\mathscr{I}^+$ would then be

$$
\ell = \partial_r , \quad n = \partial_u - \frac{1}{2}\left(1 - \frac{M}{r}\right)^2 \partial_r , \quad m = \frac{1}{r}\varepsilon_{\mathbb{S}^2}^A \partial_A , \quad \bar{m} = \frac{1}{r}\bar{\varepsilon}_{\mathbb{S}^2}^A \partial_A ,
\tag{3.8}
$$

with $\varepsilon_{\mathbb{S}^2}^A$ a complex dyad for the round 2-sphere. Charting the 2-sphere by spherical coordinates $(\theta, \phi)$, a convenient choice of this complex dyad is

$$
\varepsilon_{\mathbb{S}^2}^A \partial_A = \frac{1}{\sqrt{2}}\left(\partial_\theta + \frac{i}{\sin\theta}\partial_\phi\right) .
\tag{3.9}
$$

Using this null tetrad, the only non-zero spin coefficients, Maxwell-NP scalars and Weyl-NP

---

[23]We are using the sign convention $m \cdot \bar{m} = -\ell \cdot n = +1$, such that $g_{ab} = -2\ell_{(a}n_{b)} + 2m_{(a}\bar{m}_{b)}$.

scalars read

$$\rho_{\mathrm{NP}}^{\mathrm{ERN}} = -\frac{1}{r}, \quad \mu_{\mathrm{NP}}^{\mathrm{ERN}} = -\frac{(r-M)^2}{2r^3},$$

$$\gamma_{\mathrm{NP}}^{\mathrm{ERN}} = \frac{M(r-M)}{2r^3}, \quad \beta_{\mathrm{NP}}^{\mathrm{ERN}} = \frac{1}{2\sqrt{2}}\frac{\cot\theta}{r} = -\bar{\alpha}_{\mathrm{NP}}^{\mathrm{ERN}}, \tag{3.10}$$

$$\phi_1^{\mathrm{ERN}} = \frac{Q}{2\sqrt{4\pi}\, r^2}, \quad \Psi_2^{\mathrm{ERN}} = \frac{M(r-M)}{r^4}.$$

To study the near-$\mathscr{H}^+$ or near-$\mathscr{H}^-$ modes, we will instead use advanced or retarded Eddington-Finkelstein coordinates, $(v, \rho, x^A)$ and $(u, \rho, x^A)$, respectively, $\rho = r - M$ being the affine radial coordinate centered at the horizon,

$$ds_{\mathrm{ERN}}^2 = -\frac{\rho^2}{(M+\rho)^2}dv^2 + 2\,dv d\rho + (M+\rho)^2\,d\Omega_2^2$$

$$= -\frac{\rho^2}{(M+\rho)^2}du^2 - 2\,dud\rho + (M+\rho)^2\,d\Omega_2^2. \tag{3.11}$$

Then, a set of null tetrad vectors adapted to $\mathscr{H}^+$ would be

$$\ell = -\partial_\rho, \quad n = \partial_v + \frac{1}{2}\frac{\rho^2}{(M+\rho)^2}\partial_\rho, \quad m = \frac{1}{M+\rho}\varepsilon_{\mathbb{S}^2}^A\partial_A, \quad \bar{m} = \frac{1}{M+\rho}\bar{\varepsilon}_{\mathbb{S}^2}^A\partial_A, \tag{3.12}$$

with $\varepsilon_{\mathbb{S}^2}^A$ the same complex dyad for the 2-sphere as in Eq. (3.9), and the non-zero spin coefficients, Maxwell-NP scalars and Weyl-NP scalars are

$$\rho_{\mathrm{NP}}^{\mathrm{ERN}} = \frac{1}{M+\rho}, \quad \mu_{\mathrm{NP}}^{\mathrm{ERN}} = \frac{\rho^2}{2(M+\rho)^3},$$

$$\gamma_{\mathrm{NP}}^{\mathrm{ERN}} = -\frac{M\rho}{2(M+\rho)^3}, \quad \beta_{\mathrm{NP}}^{\mathrm{ERN}} = \frac{1}{2\sqrt{2}}\frac{\cot\theta}{M+\rho} = -\bar{\alpha}_{\mathrm{NP}}^{\mathrm{ERN}}, \tag{3.13}$$

$$\phi_1^{\mathrm{ERN}} = \frac{Q}{2\sqrt{4\pi}\,(M+\rho)^2}, \quad \Psi_2^{\mathrm{ERN}} = \frac{M\rho}{(M+\rho)^4}.$$

We announce here the change of notation compared to Section 2: here, it is the tetrad vector $n \,\hat{=}\, \partial_v$ that becomes the null normal on the horizon, rather than $\ell$. We have done this for the sole reason of presenting more compactly our succeeding analysis, such that the Couch-Torrence inversion does not change the spin-weight and boost-weight of the corresponding NP scalar.

Similarly, null tetrads adapted to the past null surfaces $\mathscr{I}^-$ or $\mathscr{H}^-$ can be obtained by the replacements $u \mapsto -v$ or $v \mapsto -u$ respectively.

The equations of motion for minimally coupled massless scalar, electromagnetic (Maxwell) and gravitational perturbations around a configuration of type-$D$ in the Petrov classification [136] were shown to acquire the following collective form within the NP formalism [137–139] (see Appendix A),

$$\Big[ (D - 2s\rho_{\mathrm{NP}} - \bar{\rho}_{\mathrm{NP}} - (2s-1)\epsilon_{\mathrm{NP}} + \bar{\epsilon}_{\mathrm{NP}})(\triangle + \mu_{\mathrm{NP}} - 2s\gamma_{\mathrm{NP}})$$

$$- (\delta - 2s\tau_{\mathrm{NP}} + \bar{\pi}_{\mathrm{NP}} - \bar{\alpha}_{\mathrm{NP}} - (2s-1)\beta_{\mathrm{NP}})(\bar{\delta} + \pi_{\mathrm{NP}} - 2s\alpha_{\mathrm{NP}})$$

$$+ (2s-1)(j-1)\Psi_2 \Big]\psi_s = 0, \tag{3.14}$$

where $s = 0$ for scalar perturbations, $s = \pm 1$ for electromagnetic perturbations and $s = \pm 2$ for gravitational perturbations. The spin-weight $s$ master variable $\psi_s$ is directly related to the

538  fundamental NP scalars according to

$$\psi_s = W^{|s|-s} \times \begin{cases} \Phi & \text{for scalar perturbations } (s=0)\,; \\ \phi_{1-s} & \text{for electromagnetic perturbations } (s=\pm 1)\,; \\ \Psi_{2-s} & \text{for gravitational perturbations } (s=\pm 2)\,, \end{cases} \qquad (3.15)$$

539  with $W$ a spin-weight zero scalar function that satisfies

$$\begin{aligned} (D-\rho_{\text{NP}})W &= 0\,, & (\delta-\tau_{\text{NP}})W &= 0\,, \\ (\triangle+\mu_{\text{NP}})W &= 0\,, & (\bar{\delta}+\pi_{\text{NP}})W &= 0\,. \end{aligned} \qquad (3.16)$$

540  For instance, if the background geometry is Ricci flat, it is typical to choose $W = \Psi_2^{1/3}$ [137,
541  138], while, if the background is an electrovacuum spacetime, one may choose $W = \phi_2^{1/2}$.
542      Let us briefly comment on the approximations involved when writing down Eq. (3.14).
543  For $s=0$, it is exact for a minimally coupled real scalar field perturbation. For $s=\pm 1$,
544  it deals with electromagnetic perturbations of an electrovacuum spacetime, but with *frozen*
545  gravitational field. The $s=\pm 2$ equation, instead, captures gravitational perturbations, but
546  with constrained electromagnetic perturbations[24]. This allowed us to set to zero all the source
547  terms that would otherwise enter in the RHS due to the coupling of electromagnetic and grav-
548  itational perturbations.
549      Focusing on the extremal Reissner-Nordström black hole background, the unified equation
550  of motion for $\psi_s$ reduces to

$$\begin{aligned} {}_{\mathscr{I}^+}\mathbb{T}_s \psi_s &= 0\,, \\ {}_{\mathscr{I}^+}\mathbb{T}_s &:= (r-M)^{-2s}\,\partial_r\,(r-M)^{2(s+1)}\,\partial_r + 2\eth'_{\mathbb{S}^2}\eth_{\mathbb{S}^2} - 2\left(r^2\partial_r + (2s+1)\,r\right)\partial_u\,, \end{aligned} \qquad (3.17)$$

551  when using the near-$\mathscr{I}$-adapted tetrad and coordinates, see Eq. (3.8), and after multiplying
552  by $-2r^2$, or to

$$\begin{aligned} {}_{\mathscr{H}^+}\mathbb{T}_s \psi_s &= 0\,, \\ {}_{\mathscr{H}^+}\mathbb{T}_s &:= \rho^{-2s}\partial_\rho\,\rho^{2(s+1)}\partial_\rho + 2\eth'_{\mathbb{S}^2}\eth_{\mathbb{S}^2} + 2\left((M+\rho)^2\,\partial_\rho + (2s+1)(M+\rho)\right)\partial_\upsilon\,, \end{aligned} \qquad (3.18)$$

553  when using the near-$\mathscr{H}$-adapted tetrad and coordinates, see Eq. (3.12), and after multiplying
554  by $-2(M+\rho)^2$. In the above expressions, $\eth_{\mathbb{S}^2}$ and $\eth'_{\mathbb{S}^2}$ are the "edth" operators on the 2-sphere,
555  which in the current spherical coordinates act on a spin-weight $s$ object according to

$$\begin{aligned} \eth_{\mathbb{S}^2} &= \frac{1}{\sqrt{2}}\left(\partial_\theta + \frac{i}{\sin\theta}\partial_\phi - s\cot\theta\right)\,, \\ \eth'_{\mathbb{S}^2} &= \frac{1}{\sqrt{2}}\left(\partial_\theta - \frac{i}{\sin\theta}\partial_\phi + s\cot\theta\right)\,, \end{aligned} \qquad (3.19)$$

556  and we have made use of the commutator $\left[\eth_{\mathbb{S}^2}, \eth'_{\mathbb{S}^2}\right] = s$. Let it also be noted that the quantity
557  $2\eth'_{\mathbb{S}^2}\eth_{\mathbb{S}^2}$ is the spin-weighted Laplace-Beltrami operator on the 2-sphere.
558      The above two operators are actually exactly the same spin-weighted wave operator, but
559  were assigned a different symbol to emphasize that they are built from coordinates and tetrad
560  vectors adapted to each null surface.

---

[24]We note here that this is not equivalent to having a frozen background electromagnetic field if the back-
ground spacetime is charged under the Maxwell field. Rather, the exact requirement for the $s=+2$ (gravitational)
equations written here, for instance, is that there exists a non-zero electromagnetic perturbation that satisfies

$$\begin{aligned} \bar{\phi}_1\Big\{&(D-\rho_{\text{NP}}+\bar{\rho}_{\text{NP}}-3\epsilon_{\text{NP}}+\bar{\epsilon}_{\text{NP}})\left[(\delta-3\tau_{\text{NP}}-2\beta_{\text{NP}})\,\phi_0^{(1)} + 2\phi_1\sigma_{\text{NP}}^{(1)}\right] \\ &+ (\delta-\tau_{\text{NP}}-\bar{\pi}_{\text{NP}}-3\beta_{\text{NP}}-\bar{\alpha}_{\text{NP}})\left[(D-3\rho_{\text{NP}}-2\epsilon_{\text{NP}})\,\phi_0^{(1)} + 2\phi_1\kappa_{\text{NP}}^{(1)}\right]\Big\} = 0\,, \end{aligned}$$

where the superscript "(1)" denotes a perturbed quantity, see e.g. the analyses of Refs. [140–143].

### 3.3 Near-$\mathscr{I}$ (Newman-Penrose) charges

Newman and Penrose famously showed the existence of an infinite tower of conserved quantities associated with linear massless fields of spacetime spin $j$ at $\mathscr{I}$ [65–67]. Remarkably, in the full (nonlinear) theory, a set of $(2j + 1)$ complex quantities remain conserved[25].

In order to extract the tower of conserved Newman-Penrose charges we first need to expand the NP scalars $\psi_s$ into near-$\mathscr{I}$ modes. We do this according to the prescription

$$\psi_s \sim \frac{1}{(r - M)^{2s+1}} \sum_{n=0}^{\infty} \frac{\psi_s^{(n)}(u, x^A)}{(r - M)^n} := \psi_s(u, r, x^A), \tag{3.20}$$

where we took into account how the peeling behavior of the fundamental NP scalars,

$$\Phi(u, r, x^A) \sim \frac{1}{r} \left[ \Phi^{(0)}(u, x^A) + \mathcal{O}(r^{-1}) \right],$$

$$\phi_{1-s}(u, r, x^A) \sim \frac{1}{r^{2+s}} \left[ \phi_{1-s}^{(0)}(u, x^A) + \mathcal{O}(r^{-1}) \right], \tag{3.21}$$

$$\Psi_{2-s}(u, r, x^A) \sim \frac{1}{r^{3+s}} \left[ \Phi_{2-s}^{(0)}(u, x^A) + \mathcal{O}(r^{-1}) \right],$$

gets translated onto the master variables $\psi_s$ and we have emphasized that the near-$\mathscr{I}$ field profile denoted by $\psi_s(u, r, x^A)$ is expected to only asymptotically converge towards the full solution $\psi_s$, hence the "$\sim$" relation.

The above near-$\mathscr{I}$ expansion is slightly different from the "canonical" prescription,

$$\psi_s(u, r, x^A) = \frac{1}{r^{2s+1}} \sum_{n=0}^{\infty} \frac{{}_{\text{can}}\psi_s^{(n)}(u, x^A)}{r^n}, \tag{3.22}$$

and can be understood as the following redefinition of the conventional near-$\mathscr{I}$ modes due to the presence of the black hole in the bulk

$$_{\text{can}}\psi_s^{(n)}(u, x^A) = \sum_{k=0}^{n} \binom{n + 2s}{k + 2s} M^{n-k} \psi_s^{(k)}(u, x^A), \tag{3.23}$$

or, inversely,

$$\psi_s^{(n)}(u, x^A) = \sum_{k=0}^{n} (-1)^{n-k} \binom{n + 2s}{k + 2s} M^{n-k} {}_{\text{can}}\psi_s^{(k)}(u, x^A). \tag{3.24}$$

In the flat limit, $M \to 0$, the two prescriptions are of course identical, but we found that this redefinition of the near-$\mathscr{I}$ modes for $M \neq 0$ actually significantly simplifies the derivation of the Newman-Penrose charges associated with spin-weight $s$ perturbations of the ERN black hole. In particular, plugging the near-$\mathscr{I}$ expansion of Eq. (3.20) into the equations of motion Eq. (3.17) outputs the following recursion relations

$$\partial_u \left( \psi_s^{(1)} + (2s + 1) M \psi_s^{(0)} \right) = -\eth'_{\mathbb{S}^2} \eth_{\mathbb{S}^2} \psi_s^{(0)}, \tag{3.25a}$$

$$\partial_u \left( (n + 1) \psi_s^{(n+1)} + (2n + 2s + 1) M \psi_s^{(n)} + (n + 2s) M \psi_s^{(n-1)} \right)$$
$$= -\left( \eth'_{\mathbb{S}^2} \eth_{\mathbb{S}^2} + \frac{1}{2} n (n + 2s + 1) \right) \psi_s^{(n)}, \tag{3.25b}$$

or, after expanding the near-$\mathscr{I}$ modes into spin-weight $s$ spherical harmonics,

$$\psi_s^{(n)}(u, x^A) = \sum_{\ell=|s|}^{\infty} \sum_{m=-\ell}^{\ell} \psi_{s\ell m}^{(n)}(u) \, {}_s Y_{\ell m}(x^A), \tag{3.26}$$

---

[25]One says that they are 'absolutely conserved' quantities.

581  to the recursion relations

$$\partial_u \left( \psi_{s\ell m}^{(1)} + (2s+1) M \psi_{s\ell m}^{(0)} \right) = \frac{1}{2} (\ell - s)(\ell + s + 1) \psi_{s\ell m}^{(0)}, \tag{3.27a}$$

$$\partial_u \left( (n+1)\psi_{s\ell m}^{(n+1)} + (2n+2s+1) M \psi_{s\ell m}^{(n)} + (n+2s) M \psi_{s\ell m}^{(n-1)} \right)$$
$$= \frac{1}{2}(\ell - s - n)(\ell + s + n + 1)\psi_{s\ell m}^{(n)}. \tag{3.27b}$$

582  From these, one directly identifies the conserved Newman-Penrose charges at level $n$ by setting
583  $\ell = s + n$,

$$_sN_{\ell m} = \psi_{s\ell m}^{(\ell - s + 1)}(u) + \frac{2\ell + 1}{\ell - s + 1} M \psi_{s\ell m}^{(\ell - s)}(u) + \frac{\ell + s}{\ell - s + 1} M^2 \psi_{s\ell m}^{(\ell - s - 1)}(u), \tag{3.28}$$
$$\Rightarrow \partial_u {}_sN_{\ell m} = 0, \quad \ell \geq |s|.$$

584  In terms of the canonical near-$\mathscr{I}$ modes ${}_{\mathrm{can}}\psi_s^{(n)}$, using the redefinition in Eq. (3.24), the
585  Newman-Penrose charges instead read

$$_sN_{\ell m} = \sum_{n=1}^{\ell - s + 1} (-1)^{\ell - s - n + 1} \frac{n}{\ell - s + 1} \binom{\ell + s}{n + 2s - 1} M^{\ell - s - n + 1} {}_{\mathrm{can}}\psi_{s\ell m}^{(n)}(u), \tag{3.29}$$

586  where ${}_{\mathrm{can}}\psi_{s\ell m}^{(n)}(u)$ are the spherical harmonic modes of ${}_{\mathrm{can}}\psi_s^{(n)}(u, x^A)$. The Newman-Penrose
587  charges derived here correctly match with previous results for scalar and electromagnetic per-
588  turbations of the ERN black hole in the current setup,[26] while they furthermore supply with
589  the Newman-Penrose charges associated with linear gravitational perturbations of the ERN
590  black hole.
591       For each value of $\ell \geq |s|$, there are $2\ell + 1$ complex charges. The first set of Newman-
592  Penrose charges corresponds to $\ell = s$ and appears only in the branch of perturbations with
593  positive spin-weight,

$$_sN_{sm} = \psi_{ssm}^{(1)}(u) + (2s+1) M \psi_{ssm}^{(0)}(u) = {}_{\mathrm{can}}\psi_{ssm}^{(1)}(u). \tag{3.30}$$

594  Their conservation turns out to be stronger than the current context of linearized perturba-
595  tions, namely, they are the $2(2s+1)$ real non-linearly conserved charges as was famously
596  demonstrated in Refs. [65, 67].
597       To make contact with the language of Ref. [65], let us now define the quantities

$$_s\mathcal{Q}_{\ell m}^{(n)}(u) := \int_{\mathbb{S}^2} d\Omega_2 \, {}_s\bar{Y}_{\ell m}(x^A) {}_{\mathrm{can}}\psi_s^{(n)}(u, x^A)$$
$$= {}_{\mathrm{can}}\psi_{s\ell m}^{(n)}(u) = \sum_{k=0}^{n} \binom{n + 2s}{k + 2s} M^{n-k} \psi_{s\ell m}^{(k)}(u), \tag{3.31}$$
$$n \geq 1, \quad |m| \leq \ell, \quad |s| \leq \ell \leq n + s - 1,$$

598  where the integration is carried over the cut-$u$ celestial sphere of $\mathscr{I}^+$. Using the recursion
599  relations for the redefined near-$\mathscr{I}$ modes, or, equivalently, plugging the canonical near-$\mathscr{I}$ ex-
600  pansion of the master variables $\psi_s$ into the equations of motion Eq. (3.17), one can show that

---

[26]For $s = 0$, Eq. (3.29) here agrees perfectly with Eq. (2.26) of Ref. [74], upon rescaling our near-$\mathscr{I}$ modes by
powers of $M$ to make all of them equi-dimensionful. For $s = +1$, working out the relation between the near-$\mathscr{I}$
modes of the Maxwell NP scalar $\phi_0$ and the near-$\mathscr{I}$ modes of the Regge-Wheeler-Zerilli master variables used by
Ref. [75], we find agreement of Eq. (3.29) here with Eq. (6.19) there.

these satisfy the evolution equations

$$
\begin{aligned}
\partial_u {}_s\mathcal{Q}_{\ell m}^{(n+1)} = {} & \frac{(\ell - s - n)(\ell + s + n + 1)}{n+1} {}_s\mathcal{Q}_{\ell m}^{(n)} \\
& + \frac{(n+s)(n+2s)}{n+1} M\, {}_s\mathcal{Q}_{\ell m}^{(n-1)} \\
& - \frac{(n+2s-1)(n+2s-1)}{2(n+1)} M^2\, {}_s\mathcal{Q}_{\ell m}^{(n-2)} \,.
\end{aligned} \tag{3.32}
$$

Compared to the original definitions of the Newman-Penrose charges for Maxwell ($s = +1$) and linearized Einstein gravity ($s = +2$) (see Eq. (3.17) and Eq. (3.31) of Ref. [66]),

$$
\left( {}_{+1}\mathcal{Q}_{\ell m}^{(n)} \right)_{\text{here}} = \left( F_m^{n-1,n-\ell} \right)_{[66]} \,, \quad \left( {}_{+2}\mathcal{Q}_{\ell m}^{(n)} \right)_{\text{here}} = \left( G_m^{n-1,n-\ell+1} \right)_{[66]} \,. \tag{3.33}
$$

The *conserved* Newman-Penrose charges are then identified with the following redefined $\mathcal{Q}$'s

$$
\begin{aligned}
{}_sN_{\ell m} = {} & \sum_{n=1}^{\ell-s+1} (-1)^{\ell-s+1-n} \frac{n}{\ell-s+1} \binom{\ell+s}{n+2s-1} M^{\ell-s+1-n}\, {}_s\mathcal{Q}_{\ell m}^{(n)}(u) \\
= {} & {}_s\mathcal{Q}_{\ell m}^{(\ell-s+1)}(u) + \mathcal{O}(M) \,, \\
& \Rightarrow \quad \partial_u {}_sN_{\ell m} = 0 \,, \quad \ell \geq |s| \,.
\end{aligned} \tag{3.34}
$$

At this point, let us comment on the appearance of Newman-Penrose charges for negative spin-weights. These Newman-Penrose charges for $s < 0$ involve sub$^{\ell+|s|+1}$-leading order near-$\mathscr{I}$ modes of the corresponding NP scalars. However, these are not new Newman-Penrose charges, on top of the ones for the branch with $s \geq 0$. Rather, the hypersurface equations of motion dictate that they are not part of the data of the problem, e.g. that they can be expressed in terms of the near-$\mathscr{I}$ modes $\left\{ \phi_{1-s}^{(0)}, \phi_0^{(n\geq 1)} \right\}$ for sourceless Maxwell fields and $\left\{ \Psi_{2-s}^{(0)}, \Psi_0^{(n\geq 1)} \right\}$ for Ricci-flat gravity. For instance, for the simple case of linearized perturbations of flat Minkowski spacetime, the hypersurface equations of motion read [66]

$$
\begin{aligned}
\frac{1}{r^{s+1}} \partial_r \left( r^{s+1} \phi_{2-s} \right) = \frac{1}{r} \eth'_{\mathbb{S}^2} \phi_{1-s} \,, \quad & 0 \leq s \leq +1 \,, \\
\frac{1}{r^{s+2}} \partial_r \left( r^{s+2} \Psi_{3-s} \right) = \frac{1}{r} \eth'_{\mathbb{S}^2} \Psi_{2-s} \,, \quad & -1 \leq s \leq +2 \,,
\end{aligned} \tag{3.35}
$$

and imply that

$$
\begin{aligned}
\phi_{1-s}^{(n\geq 1-s)} = (-1)^{1-s} \frac{(n+s-1)!}{n!} \eth'^{1-s}_{\mathbb{S}^2} \phi_0^{(n+s-1)} \,, \quad & -1 \leq s \leq 0 \,, \\
\Psi_{2-s}^{(n\geq 2-s)} = (-1)^{2-s} \frac{(n+s-2)!}{n!} \eth'^{2-s}_{\mathbb{S}^2} \Psi_0^{(n+s-2)} \,, \quad & -2 \leq s \leq +1 \,,
\end{aligned} \tag{3.36}
$$

thus showing that the subleading near-$\mathscr{I}$ modes from which the quantities ${}_{-|s|}N_{\ell m}$ are built do not carry new information.

Last, let us finish this near-$\mathscr{I}$ analysis by writing down a realization of the Newman-Penrose charges directly from asymptotic limits of transverse derivatives of the bulk field $\psi_s$. We find that

$$
{}_sN_{\ell m} = \frac{(-1)^{\ell-s+1}}{(\ell-s+1)!} \lim_{r\to\infty} \int_{\mathbb{S}^2} d\Omega_2\, {}_s\bar{Y}_{\ell m} \left[ (r-M)^2 \partial_r \right]^{\ell-s} \left[ \frac{(r-M)^{2s+1}}{r^{2s-1}} \partial_r \left( r^{2s+1} \psi_s \right) \right] \,. \tag{3.37}
$$

### 3.4 Near-$\mathscr{H}$ (Aretakis) charges

For the near-$\mathscr{H}$ charges associated with spin-weight $s$ perturbations, we follow the same procedure, that is, we first expand the NP scalar $\psi_s$ in near-$\mathscr{H}$ modes,

$$\psi_s \sim \sum_{n=0}^{\infty} \hat{\psi}_s^{(n)}(v, x^A) \left(\frac{\rho}{M}\right)^n := \hat{\psi}_s(v, \rho, x^A), \tag{3.38}$$

and then insert this into the equations of motion, Eq. (3.18), to get the recursion relations

$$M\partial_v\left(\hat{\psi}_s^{(1)} + (2s+1)\hat{\psi}_s^{(0)}\right) = -\eth'_{\mathbb{S}^2}\eth_{\mathbb{S}^2}\hat{\psi}_s^{(0)}, \tag{3.39a}$$

$$M\partial_v\left((n+1)\hat{\psi}_s^{(n+1)} + (2n+2s+1)\hat{\psi}_s^{(n)} + (n+2s)\hat{\psi}_s^{(n-1)}\right)$$
$$= -\left(\eth'_{\mathbb{S}^2}\eth_{\mathbb{S}^2} + \frac{1}{2}n(n+2s+1)\right)\hat{\psi}_s^{(n)}. \tag{3.39b}$$

We note here that we have chosen the near-$\mathscr{H}$ modes $\hat{\psi}_s^{(n)}$ to be of equal length dimension, due to the existence of the characteristic length scale $M$ provided by the black hole's size.

Proceeding, after expanding into spin-weight $s$ spherical harmonics,

$$\hat{\psi}_s^{(n)}(v, x^A) = \sum_{\ell=|s|}^{\infty} \sum_{m=-\ell}^{\ell} \hat{\psi}_{s\ell m}^{(n)}(v)\, {}_sY_{\ell m}(x^A), \tag{3.40}$$

the recursion relations reduce to

$$M\partial_v\left(\hat{\psi}_{s\ell m}^{(1)} + (2s+1)\hat{\psi}_{s\ell m}^{(0)}\right) = \frac{1}{2}(\ell-s)(\ell+s+1)\hat{\psi}_{s\ell m}^{(0)}, \tag{3.41a}$$

$$M\partial_v\left((n+1)\hat{\psi}_{s\ell m}^{(n+1)} + (2n+2s+1)\hat{\psi}_{s\ell m}^{(n)} + (n+2s)\hat{\psi}_{s\ell m}^{(n-1)}\right)$$
$$= \frac{1}{2}(\ell-s-n)(\ell+s+n+1)\hat{\psi}_{s\ell m}^{(n)}. \tag{3.41b}$$

In this form, it is straightforward to identify the conserved Aretakis charges. At level $n$, they correspond to simply setting $\ell = s + n$,

$${}_sA_{\ell m} := \hat{\psi}_{s\ell m}^{(\ell-s+1)}(v) + \frac{2\ell+1}{\ell-s+1}\hat{\psi}_{s\ell m}^{(\ell-s)}(v) + \frac{\ell+s}{\ell-s+1}\hat{\psi}_{s\ell m}^{(\ell-s-1)}(v),$$
$$\Rightarrow \quad \partial_v\, {}_sA_{\ell m} = 0, \quad \ell \geq |s|. \tag{3.42}$$

In terms of the quantities

$${}_s\hat{\mathcal{Q}}_{\ell m}^{(n)}(v) := \int_{\mathbb{S}^2} d\Omega_2\, {}_s\bar{Y}_{\ell m}(x^A)\hat{\psi}_s^{(n)}(v, x^A) = \hat{\psi}_{s\ell m}^{(n)}(v),$$
$$n \geq 1, \quad |m| \leq \ell, \quad |s| \leq \ell \leq n+s-1, \tag{3.43}$$

with the integral being taken over the cut-$v$ spherical cross-section of $\mathscr{H}^+$, the conserved Aretakis charges are identified with the following superpositions of $\hat{\mathcal{Q}}$'s

$${}_sA_{\ell m} = {}_s\hat{\mathcal{Q}}_{\ell m}^{(\ell-s+1)}(v) + \frac{2\ell+1}{\ell-s+1}\, {}_s\hat{\mathcal{Q}}_{\ell m}^{(\ell-s)}(v) + \frac{\ell+s}{\ell-s+1}\, {}_s\hat{\mathcal{Q}}_{\ell m}^{(\ell-s-1)}(v). \tag{3.44}$$

At the level of the full bulk field $\psi_s$, Eq. (3.42) is equivalent to

$${}_sA_{\ell m} = \frac{M^{\ell-s-1}}{(\ell-s+1)!}\lim_{r\to M}\int_{\mathbb{S}^2} d\Omega_2\, {}_s\bar{Y}_{\ell m}\, \partial_r^{\ell-s}\left[\frac{1}{r^{2s-1}}\partial_r\left(r^{2s+1}\psi_s\right)\right]. \tag{3.45}$$

633 For $s = 0$ or $s = +1$, our results coincide, up to overall constant factors, with the Aretakis
634 charges associated with scalar or electromagnetic (with a frozen gravitational field) pertur-
635 bations of ERN[27]. Similar to what happens with the Newman-Penrose charges of negative
636 spin-weights, the hypersurface equations imply that the Aretakis charges with $s < 0$ are also
637 dependable quantities, rather than a second infinite tower of conservation laws.

## 638  3.5  Matching of near-$\mathscr{I}$ and near-$\mathscr{H}$ charges under CT inversion

639 The curious reader might have noticed that the near-$\mathscr{H}$ recursion relations, Eq. (3.39), are
640 functionally identical to the recursion relations for the redefined near-$\mathscr{I}$ modes, Eq. (3.25).
641 This is not an accident and we will demonstrate here that it is a direct consequence of the CT
642 inversions being a conformal isometry of the ERN black hole geometry, see Eq. (3.2). Building
643 on this, we will reach the realization that the near-$\mathscr{H}$ (Aretakis) charges $_sA_{\ell m}$ and the near-$\mathscr{I}$
644 (Newman-Penrose) charges $_sN_{\ell m}$ derived above are in fact exactly equal. While this has already
645 been investigated separately for the scalar and electromagnetic cases [61, 68–71, 74, 75], the
646 present approach will allow to collectively reproduce these results and also supplement with
647 the case of gravitational perturbations, the latter requiring a more careful treatment.

648      All in all, to figure out how, for instance, the Aretakis charges get mapped under the CT
649 inversion, we need to perform a conformal transformation onto the master variables $\psi_s$ that
650 enter the equations of motion. Let us focus to the branch of perturbations with positive spin-
651 weights for the moment, for which

$$\psi_{+|s|} = \begin{cases} \Phi & \text{for } s = 0\,; \\ \phi_0 & \text{for } s = +1\,; \\ \Psi_0 & \text{for } s = +2\,. \end{cases} \tag{3.46}$$

652 For the particular conformal factor $\Omega = \frac{M}{r-M}$ associated with the CT inversion, and from the
653 fact the near-$\mathscr{H}$ tetrad vectors of Eq. (3.12) and the near-$\mathscr{I}$ tetrad vectors of Eq. (3.8) are
654 conformally related under CT inversions according to

$$\begin{pmatrix} \ell_{(3.8)} \\ n_{(3.8)} \\ m_{(3.8)} \\ \bar{m}_{(3.8)} \end{pmatrix} \xrightarrow{\text{CT}} \begin{pmatrix} \Omega^2 \ell_{(3.12)} \\ n_{(3.12)} \\ \Omega\, m_{(3.12)} \\ \Omega\, \bar{m}_{(3.12)} \end{pmatrix}, \tag{3.47}$$

655 we then have

$$\Phi\left(u, r, x^A\right) \xrightarrow{\text{CT}} \tilde{\Phi}\left(u, r, x^A\right) = \frac{M}{r-M} \hat{\Phi}\left(v \mapsto u, \rho \mapsto \frac{M^2}{r-M}, x^A\right),$$

$$\phi_0\left(u, r, x^A\right) \xrightarrow{\text{CT}} \tilde{\phi}_0\left(u, r, x^A\right) = \left(\frac{M}{r-M}\right)^3 \hat{\phi}_0\left(v \mapsto u, \rho \mapsto \frac{M^2}{r-M}, x^A\right), \tag{3.48}$$

$$\Psi_0\left(u, r, x^A\right) \xrightarrow{\text{CT}} \tilde{\Psi}_0\left(u, r, x^A\right) = \left(\frac{M}{r-M}\right)^4 \hat{\Psi}_0\left(v \mapsto u, \rho \mapsto \frac{M^2}{r-M}, x^A\right).$$

656      We now see an interesting pattern. Starting from the near-horizon expansion of the scalar
657 field $\Phi$ and the Maxwell-NP scalar $\phi_0$, the resulting CT-inverted quantities have the correct

---

[27]For the scalar Aretakis charges, see Ref. [58] and Eq. (1.16) in Ref. [144]. For the electromagnetic Aretakis
charges, after working out the relation between the Maxwell-NP scalar $\phi_0$ and the gauge field perturbation master
variables entering the Regge-Wheeler-Zerilli approach of Ref. [75], Eq. (3.42) here can be seen to match with
Eq. (6.11) there, up to an overall constant. We remind, however, that we are working in a regime where the
coupling of electromagnetic and gravitational perturbations is as explained in Footnote 24. A direct comparison
with the results of Refs. [61, 62], is therefore not straightforward.

boundary conditions near null infinity, namely, $\tilde{\Phi} \sim r^{-1}$ and $\tilde{\phi}_0 \sim r^{-3}$. Since there is only one, unique, solution of the equations of motion that satisfies the right boundary conditions at both null infinity and the horizon, this tells us that the CT-inverted quantities are in fact exactly equal to the corresponding near-$\mathscr{I}$ field profiles,

$$\tilde{\Phi}\left(u,r,x^A\right) = \Phi\left(u,r,x^A\right), \quad \tilde{\phi}_0\left(u,r,x^A\right) = \phi_0\left(u,r,x^A\right), \tag{3.49}$$

if they satisfy the same equations of motion of course. These are the types of matching conditions that are needed to explore whether the near-$\mathscr{H}$ (Aretakis) and the near-$\mathscr{I}$ (Newman-Penrose) charges coincide.

For the gravitational case, however, the CT-inverted Weyl-NP scalar $\tilde{\Psi}_0$ does *not* have the correct boundary conditions near null infinity, namely, it decays like $\sim r^{-4}$ instead of $\sim r^{-5}$. As such, the CT inverted Weyl-NP scalar is *not* expected to be the same Weyl-NP scalar one started with. In fact, the CT inverted Weyl-NP scalar does not even satisfy the same equations of motion! This is just the well-known statement that the gravitational equations of motion are *not* conformally invariant, in the sense that, for instance, Ricci flatness is not preserved under conformal transformations in four spacetime dimensions.

We will now provide a simple resolution to this complication for the gravitational case. This relies on the following remarkable property: *the spin-weighted wave operator is conformally invariant* [145]. For instance, under CT inversions, the differential operator acting on the NP scalar $\psi_s$ transforms homogeneously and with equal weights,

$$\mathscr{I}^+ \mathbb{T}_s \xrightarrow{\text{CT}} \mathscr{I}^+ \tilde{\mathbb{T}}_s = \Omega^{2s+1} \mathscr{H}^+ \mathbb{T}_s \Omega^{-2s-1}, \quad \Omega = \frac{M}{r-M}. \tag{3.50}$$

This property, however, is not special to CT inversions of ERN; it is true for any conformal transformation of the spin-weighted wave operator in Eq. (3.14) [145].

We see then that, even though the wave operator does transform homogeneously under conformal transformations, the problem resides in the fact that its conformal weights do not match with the conformal weights of the perturbations it acts on, except in the special cases $s = 0$ (scalar perturbations) and $s = +1$ (electromagnetic perturbations),

$$\mathscr{I}^+ \tilde{\mathbb{T}}_{+|s|} \begin{pmatrix} \tilde{\Phi} \\ \tilde{\phi}_0 \\ \tilde{\Psi}_0 \end{pmatrix} = \Omega^{2|s|+1} \mathscr{H}^+ \mathbb{T}_{+|s|} \begin{pmatrix} \Phi \\ \phi_0 \\ \Omega^{-1}\Psi_0 \end{pmatrix}. \tag{3.51}$$

Consequently, if $\hat{\psi}_{+|s|}\left(v,\rho,x^A\right)$ is a solution of the equations of motion in the near-horizon-adapted coordinate system, then one can reach a solution $\psi_{+|s|}\left(u,r,x^A\right)$ of the equations of motion in the near-null infinity-adapted coordinate system as follows

$$\psi_{+|s|}\left(u,r,x^A\right) = \begin{cases} \tilde{\Phi}\left(u,r,x^A\right) & \text{for } s = 0; \\ \tilde{\phi}_0\left(u,r,x^A\right) & \text{for } s = +1; \\ \Omega\tilde{\Psi}_0\left(u,r,x^A\right) & \text{for } s = +2; \end{cases} \tag{3.52}$$

$$= \left(\frac{M}{r-M}\right)^{2|s|+1} \hat{\psi}_{+|s|}\left(v \mapsto u, \rho \mapsto \frac{M^2}{r-M}, x^A\right).$$

Applying analogous arguments for the perturbations of negative spin-weights, this can be extended to the following matching statement: *if $\hat{\psi}_s\left(v,\rho,x^A\right)$ is a near-horizon expanded solution of the equations of motion, then*

$$\psi_s\left(u,r,x^A\right) = \left(\frac{M}{r-M}\right)^{2s+1} \hat{\psi}_s\left(v \mapsto u, \rho \mapsto \frac{M^2}{r-M}, x^A\right) \tag{3.53}$$

688 *is a near-null infinity expanded solution of the equations of motion.*

689     Let us now see the consequences of this matching under the CT inversion at the level of
690 the Newman-Penrose and the Aretakis charges. First of all, in terms of the asymptotic modes,
691 the matching condition tells us that

$$\text{If} \quad \hat{\psi}_s\left(v,\rho,x^A\right) = \sum_{n=0}^{\infty} \hat{\psi}_s^{(n)}\left(v,x^A\right)\left(\frac{\rho}{M}\right)^n \quad \text{and} \quad {}_{\mathscr{H}^+}\mathbb{T}_s\hat{\psi}_s = 0\,,$$

$$\text{then} \quad \psi_s\left(u,r,x^A\right) = \frac{1}{(r-M)^{2s+1}}\sum_{n=0}^{\infty}\frac{\psi_s^{(n)}\left(u,x^A\right)}{(r-M)^n} \quad \text{such that} \quad {}_{\mathscr{I}^+}\mathbb{T}_s\psi_s = 0\,,$$

692 with

$$\psi_s^{(n)}\left(u,x^A\right) = M^{n+2s+1}\hat{\psi}_s^{(n)}\left(v \mapsto u, x^A\right)\,, \tag{3.54}$$

693 or, in terms of the canonical near-$\mathscr{I}$ modes,

$$_{\text{can}}\psi_s^{(n)}\left(u,x^A\right) = M^{n+2s+1}\sum_{k=0}^{n}\binom{n+2s}{k+2s}\hat{\psi}_s^{(k)}\left(v \mapsto u, x^A\right)\,. \tag{3.55}$$

694 It is then straightforward to see that the Newman-Penrose charges exactly match the Aretakis
695 charges,

$$_sN_{\ell m} = \psi_{s\ell m}^{(\ell-s+1)(u)} + \frac{2\ell+1}{\ell-s+1}M\psi_{s\ell m}^{(\ell-s)}(u) + \frac{\ell+s}{\ell-s+1}M^2\psi_{s\ell m}^{(\ell-s-1)}(u)$$
$$= M^{\ell+s+2}\left[\hat{\psi}_{s\ell m}^{(\ell-s+1)}(v \mapsto u) + \frac{2\ell+1}{\ell-s+1}\hat{\psi}_{s\ell m}^{(\ell-s)}(v \mapsto u) + \frac{\ell+s}{\ell-s+1}\hat{\psi}_{s\ell m}^{(\ell-s-1)}(v \mapsto u)\right]\,, \tag{3.56}$$

696

$$\therefore \ _sN_{\ell m} = M^{\ell+s+2}\,{}_sA_{\ell m}\,, \quad \ell \geq |s|\,. \tag{3.57}$$

697     This result can also be seen directly from the representations of the Newman-Penrose and
698 Aretakis charges as asymptotic limits of transverse derivatives of the bulk field $\psi_s$, namely,
699 Eq. (3.37) and Eq. (3.45). Indeed, when $r \mapsto \frac{Mr}{r-M}$, the matching condition of Eq. (3.53) can
700 be seen to imply that Eq. (3.37) is equal to Eq. (3.45) according to Eq. (3.57) above.

701     The results reported here encompass in a unified manner previous results on scalar [68]
702 and electromagnetic [75] perturbations, but also extend the matching of the gravitational
703 Newman-Penrose charges[28] with the tower of conserved Aretakis charges associated with grav-
704 itational perturbations of the ERN black hole, thanks to the property of the spin-weighted wave
705 operator identified and discussed above.

## 4   The case of extremal Kerr-Newman black holes

707 We will now study an instance of a black hole geometry which is *not* self-mapped under the
708 spatial inversions of the form described in Section 2. Nevertheless, and somehow remarkably,
709 we still find that these spatial inversions allow to extract physical constraints. This is the ex-
710 ample of the extremal Kerr-Newman black hole geometry, whose horizon has the characteristic
711 feature of being twisting. As discussed in Section 2, there is no simple geometric spatial in-
712 version that conformally maps null infinity of an asymptotically flat spacetime to a twisting
713 extremal horizon at a finite distance. Despite this fact, it was demonstrated, already by Couch

---

[28]Another set of interesting gravitational charges at $\mathscr{I}$ are associated with the so-called celestial $w_{1+\infty}$ sym-
metries [40, 146–148] and were studied at finite distance in [36]. Notice, however, that these NP charges are
orthogonal to the Newman-Penrose conserved quantities (see e.g. [148, 149]).

714  & Torrence [76], that the massless Klein-Gordon equation for scalar perturbations of the ex-
715  tremal Kerr-Newman black hole still enjoys a spatial inversion symmetry, albeit one that acts
716  non-locally in coordinate space. In this section, we will extend this result to all spin-weight
717  $s$ perturbations of the rotating black hole. We will subsequently identify a sector of pertur-
718  bations onto which these spatial inversions act linearly in coordinate space and in precisely
719  such a way that one can infer an effective $\mathscr{I} \leftrightarrow \mathscr{H}$ mapping. Utilizing this, we will then
720  show that a physical consequence of these geometric spatial inversions is the exact match-
721  ing of the Newman-Penrose and Aretakis conserved quantities associated with axisymmetric
722  spin-weighted perturbations of the Kerr-Newman black hole.

## 4.1   The extremal Kerr-Newman black hole geometry

724  The Kerr-Newman black hole geometry is a solution of the electrovacuum Einstein-Maxwell
725  equations of motion. It describes an isolated, stationary and asymptotically flat black hole that
726  is, besides charged under the Maxwell field with electric charge $Q$, also rotating with angular
727  momentum $J = Ma$, $a$ being the spin parameter. An extremal Kerr-Newman (EKN) black hole
728  is one for which the gauge charges are related according to

$$a^2 + Q^2 = M^2 , \tag{4.1}$$

729  which is the condition for the event horizon to be degenerate. In Boyer-Lindquist coordinates
730  $(t, r, \theta, \phi)$, the Kerr-Newman black hole geometry is described by the line element [150, 151]

$$ds_{\text{KN}}^2 = -\frac{\Delta}{\Sigma} \left( dt - a \sin^2 \theta \, d\phi \right)^2 + \frac{\sin^2 \theta}{\Sigma} \left( a dt - \left( r^2 + a^2 \right) d\phi \right)^2 + \frac{\Sigma}{\Delta} dr^2 + \Sigma \, d\theta^2 , \tag{4.2}$$

731  where

$$\Delta = r^2 - 2Mr + a^2 + Q^2 \quad \text{and} \quad \Sigma = r^2 + a^2 \cos^2 \theta . \tag{4.3}$$

732  At extremality, the discriminant function becomes a perfect square, $\Delta = (r - M)^2$, with the
733  degenerate event horizon being located at $r = M$.

734      The Boyer-Lindquist coordinate system is singular at the event horizon. A regular coordi-
735  nate system that is adapted to a near-$\mathscr{I}^+$ or a near-$\mathscr{I}^-$ analysis is the system of retarded null
736  coordinates $(u, r, \theta, \phi_-)$ or advanced null coordinates $(v, r, \theta, \phi_+)$ respectively, related to the
737  Boyer-Lindquist coordinates according to

$$du = dt - \frac{r^2 + a^2}{\Delta} dr , \quad d\phi_- = d\phi - \frac{a}{\Delta} dr ,$$
$$dv = dt + \frac{r^2 + a^2}{\Delta} dr , \quad d\phi_+ = d\phi + \frac{a}{\Delta} dr , \tag{4.4}$$

738  In these coordinates, the Kerr-Newman metric reads

$$\begin{aligned}
ds_{\text{KN}}^2 &= -du^2 + \frac{r^2 + a^2 - \Delta}{\Sigma} \left( du - a \sin^2 \theta \, d\phi_- \right)^2 - 2 \left( du - a \sin^2 \theta \, d\phi_- \right) dr \\
&\quad + \Sigma \, d\theta^2 + \left( r^2 + a^2 \right) \sin^2 \theta \, d\phi_-^2 \\
&= -dv^2 + \frac{r^2 + a^2 - \Delta}{\Sigma} \left( dv - a \sin^2 \theta \, d\phi_+ \right)^2 + 2 \left( dv - a \sin^2 \theta \, d\phi_+ \right) dr \\
&\quad + \Sigma \, d\theta^2 + \left( r^2 + a^2 \right) \sin^2 \theta \, d\phi_+^2 .
\end{aligned} \tag{4.5}$$

739  The above two regular coordinate systems are suitable for studying observables in a near-$\mathscr{I}^+$ or
740  near-$\mathscr{I}^-$ analysis, corresponding to the limiting behavior as $r \to \infty$ while keeping $(u, \theta, \phi_-)$
741  or $(v, \theta, \phi_+)$ fixed respectively. For analogous coordinate systems that are suitable for studying

observables near the future or past event horizon one can simply employ the horizon-centered radial coordinate $\rho = r - M$. Namely, a near-$\mathcal{H}^+$ or near-$\mathcal{H}^-$ analysis corresponds to the limiting behavior as $\rho \to 0$ while keeping $(v, \theta, \phi_+)$ or $(u, \theta, \phi_-)$ fixed respectively. Let it be noted, however, that these coordinates are not null Gaussian coordinates, due to the non-vanishing of the metric components $g_{r\phi_\pm} = \pm a \sin^2 \theta$.

As opposed to ERN black hole geometry, the EKN black hole geometry does not have a simple spatial inversion conformal isometry. This should not come as a surprise though since, as we clarified in Section 2, spatial inversions of the form $r \mapsto \frac{a^2}{\rho}$ conformally map null infinity to a finite-distance extremal horizon that is *non-rotating*. As we will see shortly, however, the equations of motion for perturbations of the EKN black hole do have a spatial inversion conformal symmetry; this is a transformation that is non-local in coordinate space, but acts linearly onto the phase space of perturbations, as was already remarked for the case of minimally coupled massless scalar field perturbations in Ref. [76].

## 4.2 Equations of motion for perturbations

Let us now analyze the equations of motion governing spin-weighted perturbations of the EKN black hole. We will follow the same procedure as with the ERN black hole, namely, we will first introduce tetrad vectors adapted to each null surface of interest to subsequently extract the spin-weighted wave equation satisfied by the perturbations.

**Tetrad vectors, spin coefficients and fundamental NP scalars for near-$\mathcal{I}$ analysis**  For a near-$\mathcal{I}^+$ analysis, we choose to work with the following set of null tetrad vectors

$$\ell = \partial_r, \quad n = \frac{r^2 + a^2}{\Sigma}\left(\partial_u + \frac{a}{r^2 + a^2}\partial_{\phi_-}\right) - \frac{(r-M)^2}{2\Sigma}\partial_r,$$

$$m = \frac{1}{\sqrt{2}\,\Gamma}\left(\partial_\theta + \frac{i}{\sin\theta}\partial_{\phi_-} + ia\sin\theta\,\partial_u\right),$$

$$\bar{m} = \frac{1}{\sqrt{2}\,\bar{\Gamma}}\left(\partial_\theta - \frac{i}{\sin\theta}\partial_{\phi_-} - ia\sin\theta\,\partial_u\right),$$

(4.6)

where

$$\Gamma := r + ia\cos\theta, \tag{4.7}$$

in terms of which $\Sigma = \Gamma\bar{\Gamma}$. In the spinless limit, $a \to 0$, these reduce the tetrad vectors of Eq. (3.8) used for the ERN black hole geometry. In the black hole perturbation theory literature, these tetrad vectors are better known as the Kinnersley tetrad [152]. The Kinnersley tetrad has the property of being regular at the past event horizon but singular at the future event-horizon, as opposed to, for instance, the Hartle-Hawking tetrad [153] which is obtainable by locally boosting the Kinnersley tetrad. While this does not matter for a near-$\mathcal{I}^+$ analysis, it will be compensated in the near-$\mathcal{H}^+$ investigation by choosing an adjusted set of null tetrad vectors that is regular at $\mathcal{H}^+$.

Then, the non-zero background spin coefficients can be worked out to be

$$\rho_{\text{NP}}^{\text{EKN}} = -\frac{1}{\bar{\Gamma}}, \quad \mu_{\text{NP}}^{\text{EKN}} = -\frac{(r-M)^2}{2\Gamma\bar{\Gamma}^2},$$

$$\gamma_{\text{NP}}^{\text{EKN}} = -\frac{(r-M)^2}{2\Gamma\bar{\Gamma}^2} + \frac{r-M}{2\Gamma\bar{\Gamma}},$$

$$\tau_{\text{NP}}^{\text{EKN}} = -\frac{ia\sin\theta}{\sqrt{2}\,\Gamma\bar{\Gamma}}, \quad \pi_{\text{NP}}^{\text{EKN}} = \frac{ia\sin\theta}{\sqrt{2}\,\bar{\Gamma}^2},$$

$$\beta_{\text{NP}}^{\text{EKN}} = \frac{\cot\theta}{2\sqrt{2}\,\Gamma}, \quad \alpha_{\text{NP}}^{\text{EKN}} = -\frac{\cot\theta}{2\sqrt{2}\,\bar{\Gamma}} + \frac{ia\sin\theta}{\sqrt{2}\,\bar{\Gamma}^2},$$

(4.8)

while the only non-zero Weyl-NP and Maxwell-NP scalars are

$$\Psi_2^{\text{EKN}} = \frac{M}{\Gamma\bar{\Gamma}^3}\left(r - M + ia\cos\theta\right), \quad \phi_1^{\text{EKN}} = \frac{Q}{2\sqrt{4\pi}\,\bar{\Gamma}^2}\,. \tag{4.9}$$

**Tetrad vectors, spin coefficients and fundamental NP scalars for near-$\mathcal{H}$ analysis**   For a near-$\mathcal{H}^+$ analysis, we choose to work with the following set of null tetrad vectors

$$\begin{aligned}
\ell &= -\partial_\rho\,, \quad n = \frac{(M+\rho)^2 + a^2}{\Sigma}\left(\partial_\upsilon + \frac{a}{(M+\rho)^2 + a^2}\partial_{\phi_+}\right) + \frac{\rho^2}{2\Sigma}\partial_\rho\,, \\
m &= \frac{1}{\sqrt{2}\,\bar{\Gamma}}\left(\partial_\theta + \frac{i}{\sin\theta}\partial_{\phi_+} + ia\sin\theta\,\partial_\upsilon\right), \\
\bar{m} &= \frac{1}{\sqrt{2}\,\Gamma}\left(\partial_\theta - \frac{i}{\sin\theta}\partial_{\phi_+} - ia\sin\theta\,\partial_\upsilon\right),
\end{aligned} \tag{4.10}$$

with $\Gamma = M + \rho + ia\cos\theta$ using the current horizon-centered radial coordinate. As promised, this tetrad is regular at the future event horizon, and, hence, so will the NP scalars built from it be. The corresponding non-vanishing background spin coefficients, Weyl-NP and Maxwell-NP scalars are then given by

$$\begin{aligned}
\rho_{\text{NP}}^{\text{EKN}} &= \frac{1}{\Gamma}\,, \quad \mu_{\text{NP}}^{\text{EKN}} = +\frac{\rho^2}{2\Gamma^2\bar{\Gamma}}\,, \\
\gamma_{\text{NP}}^{\text{EKN}} &= \frac{\rho^2}{2\Gamma^2\bar{\Gamma}} - \frac{\rho}{2\Gamma\bar{\Gamma}}\,, \\
\tau_{\text{NP}}^{\text{EKN}} &= \frac{ia\sin\theta}{\sqrt{2}\,\Gamma\bar{\Gamma}}\,, \quad \pi_{\text{NP}}^{\text{EKN}} = -\frac{ia\sin\theta}{\sqrt{2}\,\Gamma^2}\,, \\
\beta_{\text{NP}}^{\text{EKN}} &= \frac{\cot\theta}{2\sqrt{2}\,\bar{\Gamma}}\,, \quad \alpha_{\text{NP}}^{\text{EKN}} = -\frac{\cot\theta}{2\sqrt{2}\,\Gamma} - \frac{ia\sin\theta}{\sqrt{2}\,\Gamma^2}\,, \\
\Psi_2^{\text{EKN}} &= \frac{M}{\Gamma^3\bar{\Gamma}}\left(\rho + ia\cos\theta\right), \quad \phi_1^{\text{EKN}} = \frac{Q}{2\sqrt{4\pi}\,\Gamma^2}\,.
\end{aligned} \tag{4.11}$$

**Teukolsky equations**   We will work again with the equations of motion of Eq. (3.14). As already remarked in Footnote 24, these are approximate for perturbations of non-zero spin-weights when the electric charge of the black hole is non-zero. They are nevertheless exact if the background electromagnetic field is absent, e.g. for the astrophysically relevant case of the electrically neutral Kerr black holes, which is also captured by our subsequent analysis.

For the $\mathscr{I}^+$-adapted tetrad vectors of Eq. (4.6) and coordinates $(u, r, \theta, \phi_-)$, and the $\mathcal{H}^+$-adapted tetrad vectors of Eq. (4.10) and coordinates $(\upsilon, \rho, \theta, \phi_+)$, the Teukolsky equations become, after multiplying Eq. (3.14) by $-2\Sigma$,

$$\mathscr{I}^+\mathbb{T}_s\psi_s = 0\,, \quad \mathscr{H}^+\mathbb{T}_s\psi_s = 0\,, \tag{4.12}$$

with

$$\begin{aligned}
\mathscr{I}^+\mathbb{T}_s &:= (r-M)^{-2s}\,\partial_r\,(r-M)^{2(s+1)}\,\partial_r - 2a\,\partial_{\phi_-}\partial_r + 2\eth_{\mathbb{S}^2}'\eth_{\mathbb{S}^2} \\
&\quad - 2\partial_u\left[\left(r^2 + a^2\right)\partial_r + (2s+1)r - \frac{1}{2}a^2\sin^2\theta\,\partial_u - a\partial_{\phi_-} + isa\cos\theta\right],
\end{aligned} \tag{4.13a}$$

$$\begin{aligned}
\mathscr{H}^+\mathbb{T}_s &:= \rho^{-2s}\partial_\rho\,\rho^{2(s+1)}\partial_\rho + 2a\,\partial_{\phi_+}\partial_\rho + 2\eth_{\mathbb{S}^2}'\eth_{\mathbb{S}^2} \\
&\quad + 2\partial_\upsilon\left[\left((M+\rho)^2 + a^2\right)\partial_\rho + (2s+1)(M+\rho) + \frac{1}{2}a^2\sin^2\theta\,\partial_\upsilon + a\partial_{\phi_+} - isa\cos\theta\right].
\end{aligned} \tag{4.13b}$$

We note here that we have isolated the purely spherical contribution to the angular operator from the remaining $\theta$-dependent pieces that enter for rotating black holes, namely, the terms $2\partial_{u/v}\left(a\partial_{\phi_{\pm}} + \frac{1}{2}a^2\sin^2\theta\,\partial_{u/v} - isa\cos\theta\right)$. These terms are the spheroidal contributions to the spin-weighted Laplace-Beltrami operator on the 2-sphere [137, 138], but the fact that they enter inside total time derivatives ensures that they will not affect our previous prescription of extracting conservation laws near $\mathscr{I}^+$ and near $\mathscr{H}^+$ by expanding into spherical harmonic modes; rather, their contribution will enter as the technical complication that the resulting charges will mix spherical harmonic modes of different orbital numbers as we will shortly see more explicitly.

## 4.3   Near-$\mathscr{I}$ (Newman-Penrose) charges

As with the ERN black hole paradigm, to extract the Newman-Penrose charges associated with spin-weight $s$ perturbations of the EKN black hole, we expand the master variables $\psi_s$ into redefined near-$\mathscr{I}$ modes

$$\psi_s \sim \frac{1}{(r-M)^{2s+1}} \sum_{n=0}^{\infty} \frac{\psi_s^{(n)}(u,\phi_-,\theta)}{(r-M)^n} := \psi_s(u,r,\phi_-,\theta)\,. \tag{4.14}$$

Inserting this near-$\mathscr{I}$ expansion into the Teukolsky equation $_{\mathscr{I}^+}\mathbb{T}_s\psi_s = 0$ gives rise to the following recursion relations

$$\begin{aligned}
\partial_u\Bigg\{ (n+1)\psi_s^{(n+1)} &+ (2n+2s+1)M\psi_s^{(n)} + (n+2s)\left(1+\chi^2\right)M^2\psi_s^{(n-1)} \\
&+ \left[\frac{1}{2}\chi^2\sin^2\theta\,M\partial_u + \chi\partial_{\phi_-} - is\chi\cos\theta\right]M\psi_s^{(n)}\Bigg\} \\
= -\left(\eth'_{\mathbb{S}^2}\eth_{\mathbb{S}^2} + \frac{1}{2}n(n+2s+1)\right)&\psi_s^{(n)} - (n+2s)\chi M\partial_{\phi_-}\psi_s^{(n-1)}\,,
\end{aligned} \tag{4.15}$$

where we have introduce the dimensionless spin parameter

$$\chi := \frac{a}{M}\,. \tag{4.16}$$

Projecting onto spin-weight $s$ spherical harmonics, this reduces to

$$\begin{aligned}
\partial_u\int_{\mathbb{S}^2} d\Omega_{2}\,_s\bar{Y}_{\ell m}\Bigg\{ (n+1)\psi_s^{(n+1)} &+ (2n+2s+1)M\psi_s^{(n)} + (n+2s)\left(1+\chi^2\right)M^2\psi_s^{(n-1)} \\
&+ \left[\frac{1}{2}\chi^2\sin^2\theta\,M\partial_u + i\chi(m-s\cos\theta)\right]M\psi_s^{(n)}\Bigg\} \\
= \int_{\mathbb{S}^2} d\Omega_{2}\,_s\bar{Y}_{\ell m}\Bigg\{ \frac{1}{2}(\ell-s-n)(\ell+s+n+1)&\psi_s^{(n)} - im\chi(n+2s)M\psi_s^{(n-1)}\Bigg\}
\end{aligned} \tag{4.17}$$

For axisymmetric ($m=0$) perturbations, in particular, one then identifies the $n$'th axisymmetric Newman-Penrose charge by setting $\ell = s+n$,

$$\begin{aligned}
_sN_{\ell,m=0} = \int_{\mathbb{S}^2} d\Omega_{2}\,_s\bar{Y}_{\ell,m=0}\Bigg\{ \\
\psi_s^{(\ell-s+1)} + \frac{2\ell+1}{\ell-s+1}M\psi_s^{(\ell-s)} &+ \frac{\ell+s}{\ell-s+1}\left(1+\chi^2\right)M^2\psi_s^{(\ell-s-1)} \\
+ \frac{1}{\ell-s+1}\left[\frac{1}{2}\chi^2\sin^2\theta\,M\partial_u - is\chi\cos\theta\right]&M\psi_s^{(\ell-s)}\Bigg\}\,,
\end{aligned} \tag{4.18}$$

$$\Rightarrow \partial_u\,_sN_{\ell,m=0} = 0\,, \quad \ell \geq |s|\,. \tag{4.19}$$

808     The first thing to observe is the qualitative new feature that the Newman-Penrose charges
809 for rotating black holes contain mixing of near-$\mathscr{I}$ spherical harmonic modes with different
810 orbital number $\ell$, due to the presence of the $\frac{1}{2}\chi^2 \sin^2\theta\, M\partial_u \psi_s^{(\ell-s)}$ and $-is\chi\cos\theta\,\psi_s^{(\ell-s)}$ terms.
811 Namely, the former term induces mixing between $\ell\pm 2$ modes, while the latter term induces
812 mixing between $\ell\pm 1$ modes. The explicit form of this mixing is written down in Appendix B.
813     The above Newman-Penrose charges are equivalent to the following asymptotic limit of
814 transverse derivatives of the bulk field $\psi_s$

$$
{}_sN_{\ell,m=0} = \frac{(-1)^{\ell-s+1}}{(\ell-s+1)!}\lim_{r\to\infty}\int_{\mathbb{S}^2} d\Omega\, {}_{2s}\bar{Y}_{\ell,m=0}\left[(r-M)^2\,\partial_r\right]^{\ell-s}\Bigg\{
$$
$$
(r-M)^{2s+1}\left[\frac{\partial_r\left[(r^2+a^2)^{\frac{2s+1}{2}}\psi_s\right]}{(r^2+a^2)^{\frac{2s-1}{2}}}-\frac{1}{2}a^2\sin^2\theta\,\partial_u\psi_s+isa\cos\theta\,\psi_s\right]\Bigg\}. \tag{4.20}
$$

815     Furthermore, for $n=0$, one can still find Newman-Penrose charges, now without the
816 restriction of the perturbations being axisymmetric. These correspond to setting $\ell=s$, which
817 occurs only in the $s\geq 0$ branch, and are given by

$$
{}_sN_{sm} = \int_{\mathbb{S}^2} d\Omega\, {}_{2s}\bar{Y}_{sm}\left\{\psi_s^{(1)}+\left[2s+1+i\chi\,(m-s\cos\theta)+\frac{1}{2}\chi^2\sin^2\theta\,M\partial_u\right]M\psi_s^{(0)}\right\}
$$
$$
= \int_{\mathbb{S}^2} d\Omega\, {}_{2s}\bar{Y}_{sm}\left\{{}_{\mathrm{can}}\psi_s^{(1)}+\left[\frac{1}{2}\chi^2\sin^2\theta\,M\partial_u+i\chi\,(m-s\cos\theta)\right]M_{\mathrm{can}}\psi_s^{(0)}\right\}, \tag{4.21}
$$

818 where in the second line we rewrote the expression in terms of the canonical near-$\mathscr{I}$ modes.
819 These are the $2s+1$ complex Newman-Penrose constants that are non-linearly conserved [65,
820 66]. One might notice that, for $\chi\neq 0$, we see additional terms as opposed to the well-known
821 result that ${}_sN_{sm} = {}_{\mathrm{can}}\psi_{ssm}^{(1)}$ [65,66]. We strongly suspect that this is related to the fact that the
822 retarded null coordinates $(u,r,\phi_-,\theta)$ we have employed are not light-cone coordinates (such
823 as null Gaussian or Bondi-like), since $g_{r\phi_-}=a\sin^2\theta\neq 0$.
824     As already mentioned, the equations of motion for the perturbations that we have used
825 are only approximate when the black hole carries an electric charge. For an extremal Kerr
826 black hole, however, for which $a^2=M^2$ and, hence, $\chi=\mathrm{sign}\{a\}:=\sigma_a$, our results are exact,
827 namely,

$$
{}_sN_{\ell,m=0}^{\mathrm{Kerr}} = \int_{\mathbb{S}^2} d\Omega\, {}_{2s}\bar{Y}_{\ell,m=0}\Bigg\{
$$
$$
\psi_s^{(\ell-s+1)}+\frac{1}{\ell-s+1}\left[2\ell+1+\frac{1}{2}\sin^2\theta\,M\partial_u-is\sigma_a\cos\theta\right]M\psi_s^{(\ell-s)} \tag{4.22}
$$
$$
+2\frac{\ell+s}{\ell-s+1}M^2\psi_s^{(\ell-s-1)}\Bigg\}.
$$

## 828  4.4  Near-$\mathscr{H}$ (Aretakis) charges

829 For the Aretakis charges investigation, we follow the analogous near-$\mathscr{H}$ procedure. We choose
830 to expand the master variables into equi-dimensionful near-$\mathscr{H}$ modes according to,

$$
\psi_s \sim \sum_{n=0}^{\infty} \hat{\psi}_s^{(n)}(v,\phi_+,\theta)\left(\frac{M\rho}{M^2+a^2}\right)^n := \hat{\psi}_s(v,\rho,\phi_+,\theta), \tag{4.23}
$$

831 namely, we chose the characteristic length dimension to be $\frac{M^2+a^2}{M}=M\left(1+\chi^2\right)$, instead of
832 just $M$. This is purely conventional and solely for future convenience, such that, when we

will spatially invert the near-$\mathscr{H}$ solution, the matching onto the redefined near-$\mathscr{I}$ modes will involve as few as possible powers of the factor $1 + \chi^2$.

Plugging this near-$\mathscr{H}$ expansion into the Teukolsky equation $_{\mathscr{H}^+}\mathbb{T}_s \psi_s = 0$, we arrive at the following recursion relations

$$
\begin{aligned}
M\partial_v \bigg\{ &(n+1)\hat{\psi}_s^{(n+1)} + (2n+2s+1)\hat{\psi}_s^{(n)} + (n+2s)\left(1+\chi^2\right)\hat{\psi}_s^{(n-1)} \\
&+ \left[\frac{1}{2}\chi^2 \sin^2\theta\, M\partial_v + \chi\partial_{\phi_+} - is\chi\cos\theta\right]\hat{\psi}_s^{(n)}\bigg\} \\
&= -\left(\eth'_{\mathbb{S}^2}\eth_{\mathbb{S}^2} + \frac{1}{2}n(n+2s+1)\right)\hat{\psi}_s^{(n)} - (n+1)\frac{\chi}{1+\chi^2}\partial_{\phi_+}\hat{\psi}_s^{(n+1)}.
\end{aligned}
\tag{4.24}
$$

A first observation here is that the terms involving $\hat{\psi}_s^{(n+1)}$ collect to form the operator $M\partial_v + \frac{\chi}{1+\chi^2}\partial_{\phi_+} = M\left(\partial_v + \frac{a}{M^2+a^2}\partial_{\phi_+}\right)$. This reflects the fact that frame-dragging effects become important on the horizon and suggests going to the co-rotating frame,

$$
\varphi_+ = \phi_+ - \frac{a}{M^2+a^2}v,
\tag{4.25}
$$

in which the recursion relations become

$$
\begin{aligned}
M\partial_v \bigg\{ &(n+1)\hat{\psi}_s^{(n+1)} + (2n+2s+1)\hat{\psi}_s^{(n)} + (n+2s)\left(1+\chi^2\right)\hat{\psi}_s^{(n-1)} \\
&+ \left[\frac{1}{2}\chi^2 \sin^2\theta\, M\partial_v + \frac{1+\chi^2\cos^2\theta}{1+\chi^2}\chi\partial_{\varphi_+} - is\chi\cos\theta\right]\hat{\psi}_s^{(n)}\bigg\} \\
&= -\left(\eth'_{\mathbb{S}^2}\eth_{\mathbb{S}^2} + \frac{1}{2}n(n+2s+1)\right)\hat{\psi}_s^{(n)} + (n+2s)\chi\partial_{\varphi_+}\hat{\psi}_s^{(n-1)} \\
&\quad + \left[2n+2s+1 - is\chi\cos\theta + \left(1 - \frac{\chi^2\sin^2\theta}{2(1+\chi^2)}\right)\chi\partial_{\varphi_+}\right]\frac{\chi}{1+\chi^2}\partial_{\varphi_+}\hat{\psi}_s^{(n)}.
\end{aligned}
\tag{4.26}
$$

As with the case of Newman-Penrose charges, projecting onto axisymmetric ($m = 0$) spin-weighted spherical harmonics,

$$
\begin{aligned}
M\partial_v \int_{\mathbb{S}^2} d\Omega\, _{2s}\bar{Y}_{\ell,m=0}\bigg\{ &(n+1)\hat{\psi}_s^{(n+1)} + (2n+2s+1)\hat{\psi}_s^{(n)} \\
&+ (n+2s)\left(1+\chi^2\right)\hat{\psi}_s^{(n-1)} + \left[\frac{1}{2}\chi^2\sin^2\theta\, M\partial_v - is\chi\cos\theta\right]\hat{\psi}_s^{(n)}\bigg\} \\
&= \int_{\mathbb{S}^2} d\Omega\, _{2s}\bar{Y}_{\ell,m=0}\left\{\frac{1}{2}(\ell-s-n)(\ell+s+n+1)\hat{\psi}_s^{(n)}\right\},
\end{aligned}
\tag{4.27}
$$

reveals that setting $\ell = s + n$ gives rise to a conservation law on the horizon, i.e. the axisymmetric Aretakis charges are given by

$$
\begin{aligned}
_s A_{\ell,m=0} = \int_{\mathbb{S}^2} d\Omega\, _{2s}\bar{Y}_{\ell,m=0}\bigg\{ & \\
\hat{\psi}_s^{(\ell-s+1)} + &\frac{2\ell+1}{\ell-s+1}\hat{\psi}_s^{(\ell-s)} + \frac{\ell+s}{\ell-s+1}\left(1+\chi^2\right)\hat{\psi}_s^{(\ell-s-1)} \\
+ &\frac{1}{\ell-s+1}\left[\frac{1}{2}\chi^2\sin^2\theta\, M\partial_v - is\chi\cos\theta\right]\hat{\psi}_s^{(\ell-s)}\bigg\},
\end{aligned}
\tag{4.28}
$$

$$
\Rightarrow \partial_v\, _s A_{\ell,m=0} = 0, \quad \ell \geq |s|.
\tag{4.29}
$$

In Appendix B, we perform explicitly the integrals to write the above Aretakis charges in terms of mixed near-$\mathscr{H}$ spherical harmonic modes.

In terms of near-$\mathcal{H}$ limits of transverse derivatives of the bulk field, this is equivalent to [60]

$$
{}_sA_{\ell,m=0} = \frac{M^{\ell-s-1}\left(1+\chi^2\right)^{\ell-s}}{(\ell-s+1)!} \lim_{r\to M} \int_{\mathbb{S}^2} d\Omega\, {}_{2\,s}\bar{Y}_{\ell,m=0}
$$

$$
\times \partial_r^{\ell-s} \left[ \frac{\partial_r\left[\left(r^2+a^2\right)^{\frac{2s+1}{2}}\psi_s\right]}{\left(r^2+a^2\right)^{\frac{2s-1}{2}}} + \frac{1}{2}a^2\sin^2\theta\,\partial_v\psi_s - isa\cos\theta\,\psi_s \right]. \tag{4.30}
$$

For an extremal Kerr black hole, for which our results become exact, the axisymmetric Aretakis charges then read

$$
{}_sA_{\ell,m=0}^{\mathrm{Kerr}} = \int_{\mathbb{S}^2} d\Omega\, {}_{2\,s}\bar{Y}_{\ell,m=0} \Bigg[
$$

$$
\hat{\psi}_s^{(\ell-s+1)} + \frac{1}{\ell-s+1}\left(2\ell+1+\frac{1}{2}\sin^2\theta\,M\partial_v - is\sigma_a\cos\theta\right)\hat{\psi}_s^{(\ell-s)} \tag{4.31}
$$

$$
+ 2\frac{\ell+s}{\ell-s+1}\hat{\psi}_s^{(\ell-s-1)} \Bigg],
$$

with the near-$\mathcal{H}$ modes defined according to the expansion

$$
\hat{\psi}_s(v,\rho,\phi_+,\theta) = \sum_{n=0}^{\infty} \hat{\psi}_s^{(n)}(v,\phi_+,\theta)\left(\frac{\rho}{2M}\right)^n, \tag{4.32}
$$

and can be seen to match with the Aretakis charges already derived in Ref. [60], up to overall normalization factors.

## 4.5   Spatial inversions symmetry of Teukolsky equations

As briefly discussed at the beginning of this section, the EKN geometry is not equipped with a conformal isometry of the form $r \mapsto \frac{\alpha^2}{\rho}$ that exchanges the null surfaces of null infinity and the event horizon, in accordance with the results of Section 2. Nevertheless, Couch & Torrence noticed that the equations of motion for minimally coupled real massless scalar perturbations of the EKN black hole do enjoy a conformal symmetry under spatial inversions of this form, but that act non-locally in coordinate space [76], a result that was recently generalized to scalar field perturbations of instances of rotating black holes in supergravity [154, 155].

Motivated by this, we will now examine the behavior of the Teukolsky operators in Eq. (4.13) under spatial inversions of the form

$$
r - M \mapsto \frac{r_{\mathrm{c}}^2}{\rho}, \quad u \mapsto v, \quad \phi_- \mapsto \phi_+. \tag{4.33}
$$

Under such inversions,

$$
{}_{\mathscr{I}^+}\mathbb{T}_s \mapsto \rho^{2s+1}{}_{\mathscr{H}^+}\mathbb{T}_s\,\rho^{-2s-1}
$$

$$
+ 2\left[\left(M^2+a^2-r_{\mathrm{c}}^2\right)\partial_v + a\partial_{\phi_+}\right]\left[\left(\frac{\rho^2}{r_{\mathrm{c}}^2}-1\right)\partial_\rho + \frac{2s+1}{\rho}\right], \tag{4.34}
$$

from which one realizes that the Teukolsky operator is conformally invariant if one formally matches the parameter $r_{\mathrm{c}}$ to be

$$
r_{\mathrm{c}}^2 = M^2 + a^2 + \frac{a\partial_{\phi_+}}{\partial_v} \quad \Rightarrow \quad {}_{\mathscr{I}^+}\mathbb{T}_s \mapsto \rho^{-2s-1}{}_{\mathscr{H}^+}\mathbb{T}_s\,\rho^{2s+1}. \tag{4.35}
$$

868  The parameter $r_c$ is non-local in coordinate space, but acts linearly in the phase space of
869  perturbations

$$r_c^2 \left( e^{-i\omega v} e^{im\phi_+} \right) = e^{-i\omega v} e^{im\phi_+} \left( M^2 + a^2 - \frac{ma}{\omega} \right), \tag{4.36}$$

870  as already demonstrated by Couch & Torrence for scalar perturbations, and here extended to
871  all spin-weight $s$ perturbations of the rotating black hole.

872       The geometric part of $r_c$ is in fact precisely what reflects the tortoise coordinate[29]

$$r_*(r) = r - M - \frac{M^2 + a^2}{r - M} + 2M \ln \frac{r - M}{\sqrt{M^2 + a^2}} = -r_* \left( M + \frac{M^2 + a^2}{r - M} \right). \tag{4.37}$$

873       The non-geometric part of the parameter $r_c$ is built from Killing vectors of the background
874  geometry, hence the no troubles when commuting it with the various metric functions. Its
875  origin can be traced back to the following single term in the Teukolsky operator,

$$\mathbb{T}_s \supset \pm 2a \partial_{\phi_\pm} \partial_r. \tag{4.38}$$

876  Evidently, this is also the term that obstructs the analytic construction of Aretakis and Newman-
877  Penrose charges for non-axisymmetric perturbations.

## 4.6   Geometric sector of spatial inversions and the matching of near-$\mathscr{I}$ and near-
##          $\mathscr{H}$ charges

880  The last technical observation around the origin of the non-local part of the spatial inver-
881  sions for spinning black holes, suggests that the sector of axisymmetric perturbations posses
882  a geometric spatial inversion conformal symmetry. Indeed, when $\partial_\phi \psi_s = 0$, the perturba-
883  tion satisfies $\mathbb{T}_s^{\mathrm{red}} \psi_s = 0$, with the following reduced Teukolsky operator adapted to each null
884  surface of interest

$$\begin{aligned}
\mathscr{I}^+ \mathbb{T}_s^{\mathrm{red}} &= (r - M)^{-2s} \partial_r (r - M)^{2(s+1)} \partial_r + 2 \eth_{\mathbb{S}^2}' \eth_{\mathbb{S}^2} \\
&\quad - 2\partial_u \left[ \left( r^2 + a^2 \right) \partial_r + (2s+1) r - \frac{1}{2} a^2 \sin^2 \theta \, \partial_u + isa \cos \theta \right],
\end{aligned} \tag{4.39a}$$

$$\begin{aligned}
\mathscr{H}^+ \mathbb{T}_s^{\mathrm{red}} &= \rho^{-2s} \partial_\rho \, \rho^{2(s+1)} \partial_\rho + 2 \eth_{\mathbb{S}^2}' \eth_{\mathbb{S}^2} \\
&\quad + 2\partial_v \left[ \left( (M+\rho)^2 + a^2 \right) \partial_\rho + (2s+1)(M+\rho) + \frac{1}{2} a^2 \sin^2 \theta \, \partial_v - isa \cos \theta \right],
\end{aligned} \tag{4.39b}$$

885  and this reduced Teukolsky operator has the advantage of precisely being conformally invariant
886  under the *geometric* spatial inversion[30]

$$\begin{aligned}
r - M &\mapsto \frac{M^2 + a^2}{\rho}, \quad u \mapsto v, \quad \phi_- \mapsto \phi_+ \\
\Rightarrow \quad \mathscr{H}^+ \mathbb{T}_s^{\mathrm{red}} &\mapsto \Omega^{-2s-1} \mathscr{I}^+ \mathbb{T}_s^{\mathrm{red}} \Omega^{2s+1}, \quad \Omega = \frac{M}{r - M},
\end{aligned} \tag{4.40}$$

---

[29]We note here that the integration constant in $r_*$ has been fixed such that

$$r_* \left( r = M + \sqrt{M^2 + a^2} \right) = 0.$$

For non-rotating (extremal Reissner-Nordström) black holes, this root at $r = 2M$ is just the location of the photon
sphere. For rotating black holes, however, this does not coincide with the photon sphere [156].

[30]We note here that the length scale entering the conformal factor, $\Omega = \frac{L}{r-M}$, can be arbitrarily chosen since the
conformal weights of the Teukolsky operator are equal. Here, we chose $L = M$ for future convenience.

a fact that was already remarked in Ref. [68] for scalar perturbations and extended here to all spin-weight $s$ perturbations.

This geometric spatial inversion acquires a very nice physical interpretation: From the fact that it acts as a reflection on the tortoise coordinate, $r_* \mapsto -r_*$, and the exchange of retarded/advanced coordinates, $u \mapsto v$ and $\phi_- \mapsto \phi_+$, one then realizes that the geometric spatial inversion exactly maps $\mathscr{I}^+$ to $\mathscr{H}^+$, and vice versa.

Using the same arguments as in Section 3.5, we then arrive to the analogous matching condition that, if $\hat{\psi}_s(v, \rho, \phi_+, \theta)$ is a near-horizon expanded solution of the reduced equations of motion, then

$$\psi_s(u, r, \phi_-, \theta) = \left(\frac{M}{r - M}\right)^{2s+1} \hat{\psi}_s\left(v \mapsto u, \rho \mapsto \frac{M^2 + a^2}{r - M}, \phi_+ \mapsto \phi_-, \theta\right) \qquad (4.41)$$

is a near-null infinity expanded solution of the reduced equations of motion. Plugging this matching condition into the Newman-Penrose charges of Eq. (4.20), and comparing with the Aretakis charges of Eq. (4.30), one then realize that these two types of charges are exactly equal, up to an overall normalization factor,

$$_sN_{\ell, m=0} = M^{\ell+s+2} {}_sA_{\ell, m=0}, \quad \ell \geq |s| \,. \qquad (4.42)$$

Equivalently, at the level of asymptotic modes, the matching condition of Eq. (4.41) tells us that, if $\hat{\psi}_s(v, \rho, \phi_+, \theta) = \sum_{n=0}^{\infty} \hat{\psi}_s^{(n)}(v, \phi_+, \theta)\left(\frac{M\rho}{M^2+a^2}\right)^n$ solves $_{\mathscr{H}^+}\mathbb{T}_s^{\text{red}}\hat{\psi}_s = 0$, then $\psi_s(u, r, \phi_-, \theta)$ $= \frac{1}{(r-M)^{2s+1}} \sum_{n=0}^{\infty} \frac{\psi_s^{(n)}(u, \phi_-, \theta)}{(r-M)^n}$ solves $_{\mathscr{I}^+}\mathbb{T}_s^{\text{red}}\psi_s = 0$, provided that

$$\psi_s^{(n)}(u, \phi_-, \theta) = M^{n+2s+1} \hat{\psi}_s^{(n)}(v \mapsto u, \phi_+ \mapsto \phi_-, \theta) \,, \qquad (4.43)$$

which precisely outputs the equality Eq. (4.42) when inserting it into the Newman-Penrose charges of Eq. (4.18) and comparing with the Aretakis charges of Eq. (4.28).

# 5 Summary and discussion

In this work, we have emphasized the existence of a conformal isomorphism between asymptotically flat spacetimes and geometries that contain an extremal, non-twisting and non-expanding horizon. The correspondence between the two types of geometries comes in the form of discrete spatial inversions that map a large distance null surface (null infinity) to a finite distance one (horizon). The conformal nature of this correspondence, nevertheless, ensures the renowned dissimilar physics near each null surface [32, 33, 115].

The geometry near the extremal horizon that corresponds to the spatially inverted asymptotically flat spacetime in general does not reside in the same asymptotically flat spacetime. A counterexample of this situation is the four-dimensional extremal Reissner-Nordström (ERN) black hole, with the spatial inversion reducing to the well-known Couch-Torrence inversion [76]. This fact allows to extract physical constraints in the form of matching conditions between quantities living on the asymptotically far null surface of null infinity and quantities living on the event horizon of the ERN black hole. We have illustrated this by further examining the relation between infinite towers of conservation laws: the near-null infinity Newman-Penrose and the near-horizon Aretakis conserved quantities. We have revisited previous analyses for scalar and electromagnetic perturbations of the ERN geometry [61, 68, 75] in a unified framework and extended these results to the more intricate case of gravitational perturbations.

We have furthermore showed that, while they seemed a priori to only lead to physical consequences for the restricted case of ERN, conformal inversions turn out to be also relevant for

the extremal Kerr-Newman (EKN) black holes. The event horizon of these black holes is now equipped with a non-zero twist and hence does not spatially invert simply to null infinity of an asymptotically flat spacetime. Despite this fact, we have demonstrated that spatial inversions of the form studied in this work still have a physical effect onto spin-weight $s$ perturbations of the EKN black hole and, in particular, impose physical constraints such as the matching of the conserved quantities.

Our results open various venues for future research, summarized below.

**More selection rules from conformal inversion**    The Aretakis and Newman-Penrose charges have previously been suggested to be associated with outgoing radiation at $\mathscr{H}^+$ and incoming radiation at $\mathscr{I}^+$ respectively [61]. However, the Couch-Torrence inversion conformal symmetry is expected to provide selection rules on other physical response properties of the black hole, namely, on its quasinormal-modes spectrum. A first hint towards this is the fact that the boundary conditions one imposes when studying the quasinormal mode spectrum of a black hole (ingoing wave at $\mathscr{H}^+$ and outgoing wave at $\mathscr{I}^+$) are preserved under the Couch-Torrence inversion. The Couch-Torrence inversion could then provide a notion of strong-weak coupling duality in which solving the perturbation equations of motion in the "strong coupling" regime of the near-zone region $\omega(r-M) \ll 1$, $\omega$ being the frequency of the perturbation, is dual to solving the perturbation equations of motion in the "weak coupling" regime $\frac{\omega M^2}{r-M} \ll 1$, and vice versa. A preliminary analysis along these lines was done in Ref. [157] which showed that the vanishing of the Love numbers associated with static scalar perturbations of the ERN black hole or static and axisymmetric scalar perturbations of the EKN black hole, follows from the Couch-Torrence inversion conformal isometry.

**Generalized Couch-Torrence inversion**    It is natural to ask whether our present approach can be applied to other examples of black holes that are equipped with a generalized Couch-Torrence (CT) inversion structure [130,131,154,155], notably, to black holes that are rotating. A significant complication that enters when the black hole is spinning is that the spatial inversions studied so far are not geometrical; rather, they act on the phase space of perturbations of the black hole, as was already remarked in the original work of Couch & Torrence [76]. In this work, we utilized the observation that the sector of axisymmetric black hole perturbations possess a generalized CT inversion is agnostic to the details of the perturbation [68]. A natural next step is to study how the phase-space spatial inversions associated with non-axisymmetric perturbations of rotating black holes restrict the physical data.

At the same time, one may wonder whether there exist spinning generalizations of the CT inversion symmetry of the ERN black hole that remain conformal isometries of the background. A first attempt along these lines could be through the Newman-Janis algorithm of constructing rotating black hole solutions, starting from a seed geometry of a non-rotating black hole [150,151,158–162]. Furthermore, for the case of EKN black holes, the observation that the parameter $r_c$ entering the generalized Couch-Torrence inversion of Eq. (4.35) involves the characteristic co-rotating operator $\partial_t + \frac{a}{M^2+a^2}\partial_\phi$ (in Boyer-Lindquist coordinates), which is also the Killing vector field that generates the event horizon of the rotating black hole, suggests that factorizing the leading order frame-dragging effects on the horizon could potentially allow to find spatial inversions that map this horizon onto null infinity and vice versa. We leave these computational prospects for near future work.

**Conformal isomorphism for twisting horizons**    Our present analysis demonstrated a conformal isomorphism between asymptotically flat spacetimes and geometries that contain an extremal, non-expanding and non-rotating horizon, by means of the spatial inversion of Eq. (2.21). One may then ask whether twisting horizons can also be incorporated withing this framework.

As demonstrated in Ref. [71], the spatial inversion presented here in general maps the geometry near an extremal and non-expanding horizon to an asymptotically flat geometry with $g_{uA} = \mathcal{O}(r)$[31]. We expect that it should be possible to generalize our analysis to this case as well.

**Self-inversion and near-horizon multipole moments**    In general, geometries that are 'self-dual' under the spatial inversions are automatically black hole geometries, since the self-inversion condition captures information about the global structure of the spacetime that contains the horizon. It would be therefore instructive to understand what are the minimum geometric conditions that eventually ensure this property. To achieve this, one would need to identify a sufficiently large class of observables constructed from the data associated with each geometry, such that these observables uniquely reconstruct the corresponding geometry.

For instance, one could identify multipole moments that live near null infinity and near the horizon and check under what conditions one can be retrieved after performing a spatial inversion on the other. While multipole moments are typically defined near spatial infinity [163–165], it was recently demonstrated in Ref. [40] that a notion of "celestial multipoles" living at null infinity plus the Newman-Penrose charges could be sufficient for this scope. On the other hand, the dynamical nature of the horizon metric obstructs the construction of multipole moments living on a generic horizon, due to the absence of a universal boundary structure at the horizon. However, non-expanding horizons appear to have sufficiently constrained dynamics to allow such a universality class of geometries to arise and, hence, attempt to define horizon multipole moments. This was partly achieved in Refs. [114, 115] which identified a set of near-horizon geometric multipole moments that uniquely characterize the intrinsic geometry of the horizon. In the same spirit, one may attempt to define horizon multipole moments associated with extremal black hole geometries using the characteristic feature of a near-horizon AdS$_2$ throat [166–168]. Another prospect would be to see how the celestial multipoles of Ref. [40] behave under spatial inversions and whether the resulting near-horizon quantities can be identified as horizon multipole moments with the expected properties. We leave these prospects for forthcoming development.

**Acknowledgments**    The authors are very grateful to James Lucietti and Simone Speziale for their comments on the manuscript. We also thank Glenn Barnich, Geoffrey Compère, Karan Fernandes, Yannick Herfray, Stefano Liberati, Giulio Neri and Jean-Philippe Nicolas for useful discussions. L.D. also thanks Gaston Giribet, Hernán González, Charles Marteau, Julio Oliva, Miguel Pino and Andrea Puhm for collaborations on horizon symmetries. S.A., P.C. and L.D. are supported by the European Research Council (ERC) Project 101076737 – CeleBH. Views and opinions expressed are however those of the authors only and do not necessarily reflect those of the European Union or the European Research Council. Neither the European Union nor the granting authority can be held responsible for them. The authors also thank the INFN Iniziativa Specifica ST&FI and the Programme "Carrollian Physics and Holography" at the Erwin-Schrödinger International Institute for Mathematics and Physics where part of this work was conducted.

## A   Elements of the Newman-Penrose formalism

In this appendix, we review basic elements of the Newman-Penrose (NP) formalism [134,135]. In the NP formalism, the spacetime metric formulation is replaced by a local frame tetrad

---

[31]Comparing with the usual Bondi fall-offs $g_{uA} = \mathcal{O}(r^0)$, this led the authors of Ref. [71] to call this geometry "weakly asymptotically flat". However, asymptotic flatness does allow for a fall-off $g_{uA} = \mathcal{O}(r^2)$, see e.g. Ref. [95].

formulation, the tetrad, in particular, being chosen to be null. The starting point is then the introduction of two real, $\{\ell, n\}$, and two complex, complex-conjugacy-related, $\{m, \bar{m}\}$, tetrad vectors, normalized as

$$\ell \cdot n = -1, \quad m \cdot \bar{m} = +1, \tag{A.1}$$

with all other inner products being zero. The metric is then reconstructed as $g_{ab} = -2\ell_{(a}n_{b)} + 2m_{(a}\bar{m}_{b)}$. The fundamental fields in the NP formalism are projections of curvature tensors onto the various null directions. More explicitly, the 6 independent components of the Maxwell field strength tensor $F_{ab}$ are repackaged into the 3 complex Maxwell-NP scalars

$$\phi_0 := F_{\ell m}, \quad \phi_1 := \frac{1}{2}\left(F_{\ell n} - F_{m\bar{m}}\right), \quad \phi_2 := F_{\bar{m}n}, \tag{A.2}$$

while the 10 independent components of the Weyl tensor $C_{abcd}$ are rearranged into the 5 complex Weyl-NP scalars

$$\begin{aligned} \Psi_0 &:= -C_{\ell m\ell m}, \quad \Psi_1 := -C_{\ell m\ell n}, \quad \Psi_2 := -C_{\ell m\bar{m}n}, \\ \Psi_3 &:= -C_{\ell n\bar{m}n}, \quad \Psi_4 := -C_{\bar{m}n\bar{m}n}, \end{aligned} \tag{A.3}$$

where we are using the shorthand notation of replacing a spacetime index with the symbol of tetrad vector it is contracted with, e.g. $F_{\ell m} := \ell^a m^b F_{ab}$.

In order to write down equations of motion in the NP formalism, one furthermore introduces the directional derivatives,

$$\begin{pmatrix} D \\ \triangle \\ \delta \\ \bar{\delta} \end{pmatrix} := \begin{pmatrix} \ell^a \\ n^a \\ m^a \\ \bar{m}^a \end{pmatrix} \nabla_a, \tag{A.4}$$

and the spacetime Christoffel symbols are traded for the 12 spin coefficients

$$\begin{aligned} \begin{pmatrix} \kappa_{\mathrm{NP}} \\ \tau_{\mathrm{NP}} \\ \sigma_{\mathrm{NP}} \\ \rho_{\mathrm{NP}} \end{pmatrix} &= -m^a \begin{pmatrix} D \\ \triangle \\ \delta \\ \bar{\delta} \end{pmatrix} \ell_a, \quad \begin{pmatrix} \pi_{\mathrm{NP}} \\ \nu_{\mathrm{NP}} \\ \mu_{\mathrm{NP}} \\ \lambda_{\mathrm{NP}} \end{pmatrix} = +\bar{m}^a \begin{pmatrix} D \\ \triangle \\ \delta \\ \bar{\delta} \end{pmatrix} n_a, \\ \begin{pmatrix} \epsilon_{\mathrm{NP}} \\ \gamma_{\mathrm{NP}} \\ \beta_{\mathrm{NP}} \\ \alpha_{\mathrm{NP}} \end{pmatrix} &:= +\frac{1}{2}\left( \bar{m}^a \begin{pmatrix} D \\ \triangle \\ \delta \\ \bar{\delta} \end{pmatrix} m_a - n^a \begin{pmatrix} D \\ \triangle \\ \delta \\ \bar{\delta} \end{pmatrix} \ell_a \right), \end{aligned} \tag{A.5}$$

where the labels "NP" have been inserted in order to avoid confusion between these NP spin coefficients and other symbols used in the current manuscript, e.g. from the symbols for the surface gravity $\kappa$ or the null Gaussian coordinate $\rho$ we employed in describing the near-horizon metric in Section 2.

## B  Newman-Penrose and Aretakis charges for extremal Kerr-Newman black holes in terms of spherical harmonic modes of perturbations

The Newman-Penrose charges associated with axisymmetric perturbations of the rotating black hole are given by Eq. (4.18). Expanding the near-$\mathscr{I}$ modes $\psi_s^{(n)}$ into spin-weighted spherical

harmonics,

$$\psi_s^{(n)}(u,\phi_-,\theta) = \sum_{\ell'=|s|}^{\infty} \sum_{m'=-\ell'}^{\ell'} \psi_{s\ell'm'}^{(n)}(u) \, {}_sY_{\ell'm'}(\phi_-,\theta) \,, \tag{B.1}$$

they reduce to

$$\begin{aligned}
{}_sN_{\ell,m=0} = {}&\psi_{s\ell,m=0}^{(\ell-s+1)} + \frac{2\ell+1}{\ell-s+1}M\psi_{s\ell,m=0}^{(\ell-s)} + \frac{\ell+s}{\ell-s+1}\left(1+\chi^2\right)M^2\psi_{s\ell,m=0}^{(\ell-s-1)} \\
&+ \frac{M}{\ell-s+1}\sum_{\ell'=|s|}^{\infty}\sum_{m'=-\ell'}^{\ell'}\left(\frac{1}{2}\chi^2\,{}_sI_{\ell\ell'm'}^{(2)}M\,\partial_u\psi_{s\ell'm'}^{(\ell-s)} - is\chi\,{}_sI_{\ell\ell'm'}^{(1)}\psi_{s\ell'm'}^{(\ell-s)}\right),
\end{aligned} \tag{B.2}$$

where

$$\begin{aligned}
{}_sI_{\ell\ell'm'}^{(1)} &:= \int_{\mathbb{S}^2} d\Omega_2 \, {}_s\bar{Y}_{\ell,m=0} \, {}_sY_{\ell'm'} \cos\theta \,, \\
{}_sI_{\ell\ell'm'}^{(2)} &:= \int_{\mathbb{S}^2} d\Omega_2 \, {}_s\bar{Y}_{\ell,m=0} \, {}_sY_{\ell'm'} \sin^2\theta \,.
\end{aligned} \tag{B.3}$$

The aim of this appendix is to compute these integrals and reveal the explicit mixing of the different $\ell$-modes induced by the non-vanishing rotation of the black hole.

The first step is to write $\cos\theta$ and $\sin^2\theta$ in the basis of spherical harmonic functions,

$$\cos\theta = \sqrt{\frac{4\pi}{3}}\,{}_0Y_{10}\,, \quad \sin^2\theta = \sqrt{\frac{16\pi}{9}}\left({}_0Y_{00} - \frac{1}{\sqrt{5}}{}_0Y_{20}\right). \tag{B.4}$$

Then, using the fact that ${}_s\bar{Y}_{\ell m} = (-1)^{s-m}\,{}_{-s}Y_{\ell,-m}$, the integrals ${}_sI_{\ell\ell'm'}^{(1)}$ and ${}_sI_{\ell\ell'm'}^{(2)}$ can be computed in terms of Wigner's $3-j$ symbols using the triple integral formula

$$\begin{aligned}
\int_{\mathbb{S}^2} d\Omega_2 \, {}_{s_1}Y_{\ell_1m_1}\,{}_{s_2}Y_{\ell_2m_2}\,{}_{s_3}Y_{\ell_3m_3} = {}&\sqrt{\frac{(2\ell_1+1)(2\ell_2+1)(2\ell_3+1)}{4\pi}} \\
&\times \begin{pmatrix} \ell_1 & \ell_2 & \ell_3 \\ m_1 & m_2 & m_3 \end{pmatrix}\begin{pmatrix} \ell_1 & \ell_2 & \ell_3 \\ -s_1 & -s_2 & -s_3 \end{pmatrix}.
\end{aligned} \tag{B.5}$$

while holds whenever $s_1 + s_2 + s_3 = 0$.

For ${}_sI_{\ell\ell'm'}^{(1)}$, this gives

$$\begin{aligned}
{}_sI_{\ell\ell'm'}^{(1)} = {}&(-1)^s\sqrt{(2\ell+1)(2\ell'+1)}\begin{pmatrix} \ell & \ell' & 1 \\ 0 & m' & 0 \end{pmatrix}\begin{pmatrix} \ell & \ell' & 1 \\ s & -s & 0 \end{pmatrix} \\
= {}&\delta_{m',0}(-1)^s\left\{\delta_{\ell',\ell-1}\sqrt{(2\ell-1)(2\ell+1)}\begin{pmatrix} \ell & \ell-1 & 1 \\ 0 & 0 & 0 \end{pmatrix}\begin{pmatrix} \ell & \ell-1 & 1 \\ s & -s & 0 \end{pmatrix}\right. \\
&\left.+ \delta_{\ell',\ell+1}\sqrt{(2\ell+1)(2\ell+3)}\begin{pmatrix} \ell & \ell+1 & 1 \\ 0 & 0 & 0 \end{pmatrix}\begin{pmatrix} \ell & \ell+1 & 1 \\ s & -s & 0 \end{pmatrix}\right\} \\
= {}&\delta_{m',0}\left\{\delta_{\ell',\ell-1}\sqrt{\frac{(\ell-s)(\ell+s)}{(2\ell-1)(2\ell+1)}} + \delta_{\ell',\ell+1}\sqrt{\frac{(\ell-s+1)(\ell+s+1)}{(2\ell+1)(2\ell+3)}}\right\},
\end{aligned} \tag{B.6}$$

where in the second line we applied the selection rules imposed by Wigner's $3-j$ symbols and in the third line we wrote their explicit values. Following the same procedure for ${}_sI_{\ell\ell'm'}^{(2)}$, we

1050 find

$$
{}_sI^{(2)}_{\ell\ell'm'} = \delta_{m',0}\Bigg\{ -\delta_{\ell',\ell-2}\sqrt{\frac{(\ell-s-1)(\ell+s-1)(\ell-s)(\ell+s)}{(2\ell-3)(2\ell-1)^2(2\ell+1)}}
$$
$$
+ \delta_{\ell',\ell}\frac{2}{3}\left[1+\sqrt{\frac{\ell(\ell+1)}{(2\ell-1)(2\ell+3)}}\right] \tag{B.7}
$$
$$
- \delta_{\ell',\ell+2}\sqrt{\frac{(\ell-s+1)(\ell+s+1)(\ell-s+2)(\ell+s+2)}{(2\ell+1)(2\ell+3)^2(2\ell+5)}}\Bigg\}.
$$

1051 In summary, putting everything together, if we expand the perturbation $\psi_s$ into near-$\mathscr{I}$
1052 spherical harmonic modes according to

$$
\psi_s(u,r,\phi_-,\theta) = \frac{1}{(r-M)^{2s+1}}\sum_{n=0}^{\infty}\sum_{\ell=|s|}^{\infty}\sum_{m=-\ell}^{\ell}\frac{\psi^{(n)}_{s\ell m}(u)}{(r-M)^n}\,{}_sY_{\ell m}(\phi_-,\theta)\,, \tag{B.8}
$$

1053 then the axisymmetric Newman-Penrose charges of Eq. (4.20) have the following explicit form

$$
{}_sN_{\ell,m=0} = \psi^{(\ell-s+1)}_{s\ell,m=0} + \frac{2\ell+1}{\ell-s+1}M\psi^{(\ell-s)}_{s\ell,m=0} + \frac{\ell+s}{\ell-s+1}\left(1+\chi^2\right)M^2\psi^{(\ell-s-1)}_{s\ell,m=0}
$$
$$
-\frac{M}{\ell-s+1}\Bigg\{\frac{\chi^2}{2}\sqrt{\frac{(\ell-s-1)(\ell+s-1)(\ell-s)(\ell+s)}{(2\ell-3)(2\ell-1)^2(2\ell+1)}}M\partial_u\psi^{(\ell-s)}_{s,\ell-2,m=0}
$$
$$
+is\chi\sqrt{\frac{(\ell-s)(\ell+s)}{(2\ell-1)(2\ell+1)}}\psi^{(\ell-s)}_{s,\ell-1,m=0} - \frac{\chi^2}{3}\left[1+\sqrt{\frac{\ell(\ell+1)}{(2\ell-1)(2\ell+3)}}\right]M\partial_u\psi^{(\ell-s)}_{s\ell,m=0} \tag{B.9}
$$
$$
+is\chi\sqrt{\frac{(\ell-s+1)(\ell+s+1)}{(2\ell+1)(2\ell+3)}}\psi^{(\ell-s)}_{s,\ell+1,m=0}
$$
$$
+\frac{\chi^2}{2}\sqrt{\frac{(\ell-s+1)(\ell+s+1)(\ell-s+2)(\ell+s+2)}{(2\ell+1)(2\ell+3)^2(2\ell+5)}}M\partial_u\psi^{(\ell-s)}_{s,\ell+2,m=0}\Bigg\}.
$$

1054 Similarly, if we expand the perturbation $\psi_s$ into near-$\mathscr{H}$ spherical harmonic modes ac-
1055 cording to

$$
\hat{\psi}_s(v,\rho,\phi_+,\theta) = \sum_{n=0}^{\infty}\sum_{\ell=|s|}^{\infty}\sum_{m=-\ell}^{\ell}\hat{\psi}^{(n)}_{s\ell m}(u)\left(\frac{M\rho}{M^2+a^2}\right)^n\,{}_sY_{\ell m}(\phi_+,\theta)\,, \tag{B.10}
$$

1056 then the axisymmetric Aretakis charges of Eq. (4.30) read

$$
{}_sA_{\ell,m=0} = \hat{\psi}^{(\ell-s+1)}_{s\ell,m=0} + \frac{2\ell+1}{\ell-s+1}\hat{\psi}^{(\ell-s)}_{s\ell,m=0} + \frac{\ell+s}{\ell-s+1}\left(1+\chi^2\right)\hat{\psi}^{(\ell-s-1)}_{s\ell,m=0}
$$
$$
-\frac{1}{\ell-s+1}\Bigg\{\frac{\chi^2}{2}\sqrt{\frac{(\ell-s-1)(\ell+s-1)(\ell-s)(\ell+s)}{(2\ell-3)(2\ell-1)^2(2\ell+1)}}M\partial_v\hat{\psi}^{(\ell-s)}_{s,\ell-2,m=0}
$$
$$
+is\chi\sqrt{\frac{(\ell-s)(\ell+s)}{(2\ell-1)(2\ell+1)}}\hat{\psi}^{(\ell-s)}_{s,\ell-1,m=0} - \frac{\chi^2}{3}\left[1+\sqrt{\frac{\ell(\ell+1)}{(2\ell-1)(2\ell+3)}}\right]M\partial_v\hat{\psi}^{(\ell-s)}_{s\ell,m=0} \tag{B.11}
$$
$$
+is\chi\sqrt{\frac{(\ell-s+1)(\ell+s+1)}{(2\ell+1)(2\ell+3)}}\hat{\psi}^{(\ell-s)}_{s,\ell+1,m=0}
$$
$$
+\frac{\chi^2}{2}\sqrt{\frac{(\ell-s+1)(\ell+s+1)(\ell-s+2)(\ell+s+2)}{(2\ell+1)(2\ell+3)^2(2\ell+5)}}M\partial_v\hat{\psi}^{(\ell-s)}_{s,\ell+2,m=0}\Bigg\}.
$$

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
