# Peer review of "Null infinity as an inverted extremal horizon: Matching an infinite set of conserved quantities for gravitational perturbations"

_SciPost Physics_

## Round 1 · Referee Report · Simone Speziale (Referee 1) · 2025-9-11

Report

Dear editor,

first of all my apologies to the authors for the delay of my report.

The paper studies the relation between future null infinity and certain physical horizons. It extends previous results in a non trivial way, and I am very happy to recommend it for publication. It does a nice job of reviewing the existing literature and results, including comparing notations, and of explaining in which new directions they are extending them, and their motivations. Among the results, I find particularly remarkable the matching of charges for higher spins described at the end of section 3, and in section 4 the extension of the already surprising result of Couch and Torrence in mapping scalar perturbation, to higher spins.

I have some small suggestions for optional amendments:

p1. It may be useful to stress in the introduction that the dictionary set up in this paper is based on divergence-free conformal frames, namely on describing null infinity as surface with vanishing expansion

Footnote 1: i find the term "dynamical event horizons” confusing, usually these two adjectives are opposite of one another: an event horizon is global, teleological and completely stationary, hence non dynamical. Maybe the authors could explain better what they mean?

p3: I think it would be useful to include the references 117-120 with 44-52 at the end of the paragraph; and also to say explicitly that the key difference between the full NU group (which the authors use) and the smaller group considered in [117-120] (which is the analogue of the BMSW group at null infinity) is due to adding the inaffinity of the arbitrary null normal to the universal structure, a possibility already pointed out in [45]. I would also suggest adding the reference https://inspirehep.net/literature/2641560 to that list of relevant works for the phase space on null hypersurfaces.

In the fourth paragraph of that page, the authors talk about Aretakis conserved quantities, but their existence has not been mentioned yet, so the sentence is a bit out of context.

p.10 : I would add in the first sentence of 2.3 the specification “…whose boundary is scri **in a divergence-free completion**, is diffeomorphic…”

Below 2.21, \alpha (an arbitrary real function at this stage I suppose? can it be time dependent? or a constant only?) is not defined

Page 12, Footnote 15: Among the references given, [36] seems to contain a similar idea of mapping between null infinity and horizons, and investigating sub-leading charges. If the authors know the differences/similarities in scopes and results, it would be useful to the reader to comment on them.

p13: in reference to my comment earlier, and given the attention the authors are giving to comparing the literature, it may be useful to add below 2.31 that the case considered in 117 corresponds to n=2 (this characterization is only valid at k=0, but this is anyways the context of the present paper)

p.27: is the lack of self-mapping in the extremal KN case ultimately due to the fact that the Hajicek 1-form vanishes in one case but not the other? if yes, it may be useful to add this comment

Recommendation

Publish (easily meets expectations and criteria for this Journal; among top 50%)

  • validity: -
  • significance: -
  • originality: -
  • clarity: -
  • formatting: -
  • grammar: -

Author:  Laura Donnay  on 2025-12-19  [id 6161]

(in reply to Report 1 by Simone Speziale on 2025-09-11)
Category:
answer to question
correction

We would like to thank the Referee for their careful reading of the manuscript and their comments and suggestions to improve the manuscript.

p1. We have added a footnote (footnote 5) in the introduction to emphasize that $\mathscr I$ is a NEH that is furthermore extremal and non-twisting in the preferred divergence-free conformal frames [32,33].

We have corrected footnote 1 (removed `dynamical' as indeed we wanted to emphasize the global properties in this sentence).

We have included the references [117, 119,120] with Refs. [44-52] at the end of the paragraph. Also, we have added Ref. https://arxiv.org/abs/2303.07262 there.
We have also rephrased the sentence in the third paragraph when we talk about the Aretakis conserved quantities to make it clearer.

We have clarified that we meant in the sense as defined previously (namely with a choice of divergence-free conformal frames).

We have added a definition for $\alpha$ after (2.21).

About the comparison with [36]: the gravitational charges considered in [36] are associated with the so-called celestial $w_{1+\infty}$ symmetries. The latter are orthogonal to the set of NP conserved quantities at null infinity, which are the ones considered in this work (see e.g. [147, 148]). We have added a footnote on p.33 to clarify that.

We pointed out that Refs [45,52] correspond to the case $n=2$ below eq. (2.31).

We specified that the absence for a self-dual map for EKN is due to a non-vanishing twist.

---

## Round 1 · Referee Report · Anonymous (Referee 3) · 2025-12-8

Report

The authors extend previous work on mapping null infinity to the black hole horizon through a spatial inversion. They provide a uniform treatment and generalize the construction to higher-spin perturbations. Overall, I found the paper well written and technically sound. I am happy to recommend it for publication.

I have only a few remaining questions and comments that should be addressed:
• I agree with Referee 1 that a clarification regarding the difference/comparaison with Ref. [36] is needed.
• Line 282: When the authors refer to the Bondi news tensor, they provide the expression for the conformal metric on the 2-sphere using the Geroch tensor. However, the Geroch tensor is defined for any spherical metric; a brief clarification here would be helpful.
• Line 326: The authors choose to fix q_{ab} the celestial metric. Could they explain the motivation for this choice? Or would relaxation the condition on the metric q_{ab} yield more general class of solutions under the spatial inversion?
• Line 378: Please define \Sigma
• Line 386: I suggest adding T next to “supertranslation,” since it was not previously stated that these were the supertranslations.
• Line 901: The equation runs off the page; formatting should be corrected.
• Line 974: I am not sure I agree with the comment made in that line. You can have asymptotic flat spacetime (simple in the sense of Penrose) with g_{uA} of order r, even r^2 is possible. For that it is enough to allow for $\beta_0$ (leading term in g_ur) and $U_0^A$ (leading term in U^A) respectively. Why the authors choose to implement via log (r) term in g_(AB}? Moreover it is not clear whether their construction will hold as imposing the equations of motion might not yield a consistent solution space.
• In the table of content, the entry ``references’’ appears twice.

Recommendation

Publish (easily meets expectations and criteria for this Journal; among top 50%)

---

## Round 1 · Referee Report · Massimo Bianchi (Referee 2) · 2025-12-8

Strengths

1- Clarity 2- Originality 3- Correctness 4-Precision 5- Timeliness

Weaknesses

None

Report

The paper "Null infinity as an inverted extremal horizon: Matching an
infinite set of conserved quantities for gravitational perturbations" is
very interesting, original and far-reaching. As suggested by the title,
the authors establish a robust duality between extremal non-expanding
horizons and null infinity, based of the conformal inversions originally
proposed by Couch and Torrence. In particular they check the validity of
the duality by matching near-infinity Newman-Penrose charges with
near-horizon Aretakis charges and study in great detail two prototypical
'self-dual' (that they call `self-inverted') examples: the extremal Reissner-Nordström black hole, whereby Couch-Torrence inversions preserve the metric up to a Weyl rescaling, and the extremal Kerr-Newman black hole geometry, whereby Couch-Torrence inversions do not preserve the metric up to a Weyl rescaling but act as symmetries of the (linearized) field equations for massless perturbations. The paper is organized in five Sections and 2 appendices. After a very clear and motivating introduction, Section 2 is devoted to a detailed discussion of the duality between null infinity and extremal non-expanding horizons. In addition to a very pedagogical presentation of the geometry and the (linearized) symmetries, the connection between Newman-Penrose charges and Aretakis charges is spelled out. The` self-dual' (`self-inverted') case of the extremal
Reissner-Nordström
black hole is discussed in a fully exhaustive fashion in Section 3. In
Section 4 the authors analyze extremal Kerr-Newman black holes and
relying on Newman-Penrose formalism and Teukolsky equations manage to
match near-infinity and
near-horizon charges even for fields with spin (vector and tensor
perturbations).
Conclusions and a line for future investigation are drawn in Section 5,
while appendix A offers a useful introduction to the Newman-Penrose
formalism and
appendix B contains cumbersome expressions for the N-P and A cahrges of
extremal Kerr-Newman black holes in terms of spherical harmonics.
The bibliography is rather complete and useful both for original and up
to date references. The presentation is extremely well organized and
pleasant to read. There are two points the authors may desire to address
or simply comment on. The first is the generalization to the non-linear
level, whereby it is known that only 10 Newman-Penrose charges survive.
The second consist in the role of the light-ring in the self-dual case,
eg extremal RN and the 'fate' of the light-rings in the case of Kerr
(and Kerr-Newman) BHs.

Requested changes

Minor optional additional comments 1- Generalization to non-linear level 2-Role of the light-ring in the self-dual case

Recommendation

Publish (surpasses expectations and criteria for this Journal; among top 10%)

  • validity: top
  • significance: top
  • originality: high
  • clarity: top
  • formatting: perfect
  • grammar: perfect

Author:  Laura Donnay  on 2025-12-19  [id 6162]

(in reply to Report 2 by Massimo Bianchi on 2025-12-08)

We want the thank the Referee for his reading and interest in our work.

The generalization to the non-linear level is an interesting question, given indeed the presence of 10 non-linearly conserved NP quantities. Regarding this, let us point out that it is not clear whether there exist Aretakis conserved quantities associated with non-linear perturbations of ERN. Such perturbations were analyzed numerically in https://arxiv.org/abs/1307.6800 and there it was argued that the generic end-point of the instability is a non-extreme solution. Additionally, the non-linear geometry is not self-inverted anymore, that is, there are no conventional CT inversion conformal symmetries for the non-extremal Reissner-Nordstr\"om black hole.

We believe that the geodesics point of view of the CT inversions, which keeps fixed the photon sphere, could guide potential generalizations of these types of discrete conformal symmetries, a direction we are currently investigating. We have added a footnote (footnote 21) to emphasize this point, together with the fact that it has also been utilized in Refs. [129, 130] to study physical implications on geodesic observables.

---

## Editorial Decision

awaiting_resubmission